# FTIR-derived soil degradation indices and stochastic modelling of organic matter–sediment dynamics in a Mediterranean watershed: A Northern Apennines case study

Manuel La Licata[1,2,3]*, Odunayo D. Adeniyi[2,4], Ruth H. Ellerbrock[5], Nisha Bhattarai[5], Alberto Bosino[6], Natalie Papke[1], Jörg Schaller[5], Michael Maerker[1,2,3]

**1** Working Group on Soil Erosion and Feedbacks, Landscape Functioning, Leibniz Centre for Agricultural Landscape Research (ZALF), Müncheberg, Germany, **2** Department of Earth and Environmental Sciences, University of Pavia, Pavia, Italy, **3** Institute of Geosciences and Earth Resources, National Research Council of Italy, Pavia, Italy, **4** CMCC Foundation, Euro-Mediterranean Centre on Climate Change, Milan, Italy, **5** Working Group on Silicon Biogeochemistry, Landscape Functioning, Leibniz Centre for Agricultural Landscape Research (ZALF), Müncheberg, Germany, **6** Department of Earth and Environmental Sciences, University of Milano-Bicocca, Milan, Italy

* manuel.lalicata@zalf.de

## Abstract

In this study we explored the relationships between Soil Organic Matter (SOM) properties, serving as potential indicators of soil degradation and erosion, and environmental, geomorphic, and hydrological characteristics in an agricultural-forested Mediterranean watershed. SOM composition of fluvial sediments sampled across the watershed was analysed using FTIR spectroscopy to calculate FTIR-based proxies for the relative hydrophobicity of SOM, Cation Exchange Capacity (CEC), and organic-matter-cation associations. To investigate geospatial relationships between SOM composition influencing erosion susceptibility and the factors driving its variability at the watershed scale, such as terrain characteristics, soil properties, lithological, and LULC data, we used a Random Forest modelling approach. Our findings indicate that the size and configuration of the contributing areas associated with the sampling points played a crucial role in interpreting the relationships between SOM composition and environmental factors. Oak, hornbeam, and chestnut forests influence hydrophobic organic matter accumulation, making soils more prone to water erosion, where clay content potentially intensifies erosion susceptibility under particular climatic conditions. Moreover, SOM chemical components were spatially linked to sediment dynamics and organic matter connectivity across the watershed, with topographic features such as elevation and channel network base level being key factors. Also, CEC was found to be a potential indicator of soil erosion in geomorphologically active areas. Lastly, carbonate-rich soils appeared to positively influence organic matter-cation associations, potentially enhancing aggregate stability and reducing erosion susceptibility. This study provides significant new insights into the complex

**Data availability statement:** The datasets related to the three experiments, along with the related raster and shapefile data, contributing areas, and a shapefile containing the location of the sampling points, are available at the following link: https://doi.org/10.5281/zenodo.15097417. The code of the Random Forest model implemented in this study is available at the following link: https://github.com/Odunayo3/SoilDegradationModeling.git. Additional information and outputs are available along with the Supporting Information files.

**Funding:** This research was conducted with the financial support of the Earth and Environmental Sciences PhD-PON program (Research & Innovation, 2014e2020, Education and research for recovery - REACT-EU, DOT1322534-4) of University of Pavia, Department of Earth and Environmental Sciences. This research was also funded by the PRIN 2022 project by the Italian Ministry of University and Research, entitled: "Full cOveRage, Multi-scAle and multi-sensor geomorphological map: a practical tool for TerrItOrial plaNning – FORMATION" (2022C2XPK7_004). This research was also supported by the Polish National Agency for Academic Exchange - Urgency Grants program project contract no. BPN/GIN/2024/1/00008. The funders had no role in study design, data collection and analysis, decision to publish, or preparation of the manuscript.

**Competing interests:** The authors have declared that no competing interests exist.

**Abbreviations:** ALE, Accumulated Local Effects; CA, Contributing Area; CARG, Italian Official Geological Cartography; CCC, Concordance Correlation Coefficient; CEC, Cation Exchange Capacity; CNBL, Channel Network Base Level; DDC, Downslope Distance Gradient; DOM, Dissolved Organic Matter; DTM, Digital Terrain Model; FTIR, Fourier Transform Infrared; GIS, Geographic Information System; HOTSED, Hotspots of sediment sources and related dynamics; HPD, Hazard Map of sediment Production and Delivery; IM, Inventory Map; LOOCV, Leave-One-Out Cross-Validation; LULC, Land Use/Land Cover; MSE, Mean Squared Error; NSE, Nash-Sutcliffe Efficiency; RER, Emilia-Romagna Region; RF, Random Forest; RMSE, Root Mean Squared Error; SOC, Soil Organic Carbon; SOM, Soil Organic Matter; STD, Standard Deviation; VDCN, Vertical Distance to Channel Network; WN, Wavenumber.

relationships between SOM composition, environmental predictors, and soil erosion in Mediterranean watersheds, supporting novel research hypotheses and perspectives from both a scientific and applicative point of view.

---

## 1 Introduction

Soil is a vital resource that sustains life by facilitating the exchange of mass, energy, and biodiversity [1]. It plays an essential role in supporting ecosystems by processing inputs and generating outputs that influence groundwater, vegetation, atmosphere, and surface waters [2,3]. However, soils are essentially non-renewable on human timescales, making them vulnerable to degradation due to complex interactions occurring across various spatial and temporal scales [4]. Soil degradation, exacerbated by human activities and climate changes, is a significant threat to social-ecological landscapes, particularly through soil erosion, which undermines land productivity, ecosystem services, and socio-economic stability [5–7]. Some regions such as the Mediterranean are more vulnerable to land degradation triggered by geomorphic activity, particularly due to pronounced seasonal climatic variability, steep slopes, and a long history of human pressure [8–10]. Nonetheless, proper management practices offer the potential for soil restoration and recovery, highlighting the importance of understanding the relationships between soil functions, degradation, and resilience [11], particularly in landscapes with complex nature-human interactions [12].

Soil Organic Matter (SOM) is a key factor influencing soil response to external forcings, playing a critical role in maintaining soil quality and health [13]. Moreover, it influences a range of physical, chemical, and biological properties [14,15], such as aggregate stability, bulk density, water-holding capacity, erodibility, and compaction, as well as nutrient levels, Cation Exchange Capacity (CEC), and pH [16,17]. Many SOM-related properties are indeed used as indicators of soil degradation susceptibility [4], particularly to water erosion [18,19].

SOM plays a crucial role in influencing soil erodibility [20,21], which is essential in regions vulnerable to erosion like the Mediterranean. However, the relationship between SOM and erosion processes is complex and multifaceted, as SOM's interactions with soil minerals and water infiltration behaviour can either enhance or reduce erosion. Thus, SOM chemical composition is critical in determining soil structure, as well as its response to water infiltration and overland flow [19].

For instance, hydrophobic organic compounds typically reduce water infiltration capacity with the consequence increase in surface runoff [22], thereby intensifying soil erosion in certain conditions [23]—such as after prolonged dry periods followed by heavy rainfall or in areas with limited vegetation cover and variable soil moisture, high slope gradients, and affected by wildfires [e.g., 21,24–28]. These conditions are typical of Mediterranean-type ecosystems [29]. However, the impact of hydrophobicity on water-induced erodibility depends on the balance between hydrophobic and hydrophilic components [30,31] and varies with environmental conditions and soil characteristics [32–35]. Additional SOM-related properties, such as CEC and organic

matter-cation associations [36,37], can also serve as indicators of land degradation processes. High levels of these properties improve soil aggregate stability, increasing resistance to disintegration under erosive forces [19]. Anyway, the effect of SOM to these properties is complex and depends on several factors, e.g., chemical SOM characteristics, pH, soil texture, clay minerals, and pedogenic oxides [38–40].

However, the watershed-scale relationships between SOM chemical characteristics, environmental features, and landscape connectivity, along with their impact on organic matter depletion, redistribution, and sediment dynamics, remain relatively underexplored. Addressing these relationships is essential for understanding the role of erosion-sedimentation patterns in carbon cycling and sequestration [41]. Recent studies underscore the need to investigate the chemical properties and spatial distribution of transported organic matter, since it varies significantly due to factors such as land use, altitude, clay content, pH, human disturbance, and hydrological connectivity [42,43]. Additionally, studies focusing on the effects of climatic events, such as rainstorms, on Dissolved Organic Matter (DOM) composition and related export dynamics highlight the implications for water quality protection and pollutant contamination [44,45]. Such interactions in geomorphologically dynamic and human-altered Mediterranean environments remain poorly studied. This emphasizes the need for interdisciplinary studies investigating the interplay between SOM characteristics and environmental factors for understanding watershed system functioning in soil degradation-prone environments [46].

This study aims to propose a novel methodological framework to investigate the geospatial relationships between SOM properties, serving as potential indicators of soil degradation and erosion, and the environmental, geomorphological, and hydrological characteristics in a Mediterranean watershed. The upper Val d'Arda (N-Apennines, Italy) was selected as the study area, representing an intensely altered agricultural-forested temperate Mediterranean watershed with a highly dynamic geomorphological setting [47].

According to Johnston & Aochi [48] and Margenot et al. [49], Fourier-transform infrared (FTIR) spectroscopy can be employed to analyse SOM composition, focusing on specific functional groups and their interactions with cations. This technique is particularly suitable for its rapidity, cost-effectiveness, non-destructive nature, and minimal sample requirements [49]. FTIR spectroscopy enables the analysis of SOM organic constituents by examining absorption bands, such as hydrophobic alkyl ($CH$, $CH_2$, $CH_3$) groups, hydrophilic carbonyl and carboxyl ($C=O$) groups, hydroxyl ($OH$) groups, and polysaccharide ($C–O–C$) groups [50]. By relating the intensities of these absorption bands in ratios (i.e., FTIR indices), it is possible to derive useful information on SOM composition, including relative hydrophobic group content [indicating the potential wettability of SOM; [51,52] and CEC [39,50]. Moreover, interactions between organic matter and polyvalent cations can be assessed by analysing an absorption band linked to both relative carbonate content and the $COO^-$ functional group [53–55].

Thus, to assess the relationships between SOM composition influencing erosion susceptibility and the environmental factors driving its variability at the watershed scale, FTIR indices derived from fluvial sediment samples collected across the watershed were integrated with terrain, soil properties, lithological, and land use data using a Random Forest (RF) machine learning model [56,57]. This integration aimed to uncover new insights into erosion processes, sediment dynamics, and organic matter connectivity at watershed scale. This study, which is novel in its holistic and interdisciplinary approach, not only builds on existing research by linking SOM composition with soil erosion, but also offers a new methodological approach for understanding these processes in complex Mediterranean landscapes. The findings contribute to the broader field of soil conservation by emphasizing the critical role of SOM characterization in assessing soil erosion and organic matter dynamics, offering significant implications for socio-ecological landscapes, where soil erosion and degradation assessments are critical challenges.

## 2 Study area

The upper Val d'Arda is an agricultural-forested watershed located in the western Emilia-Romagna region of the Northern Apennines (Emilian Apennines), Italy (Fig 1a). The watershed spans approximately 14 km from southwest to northeast

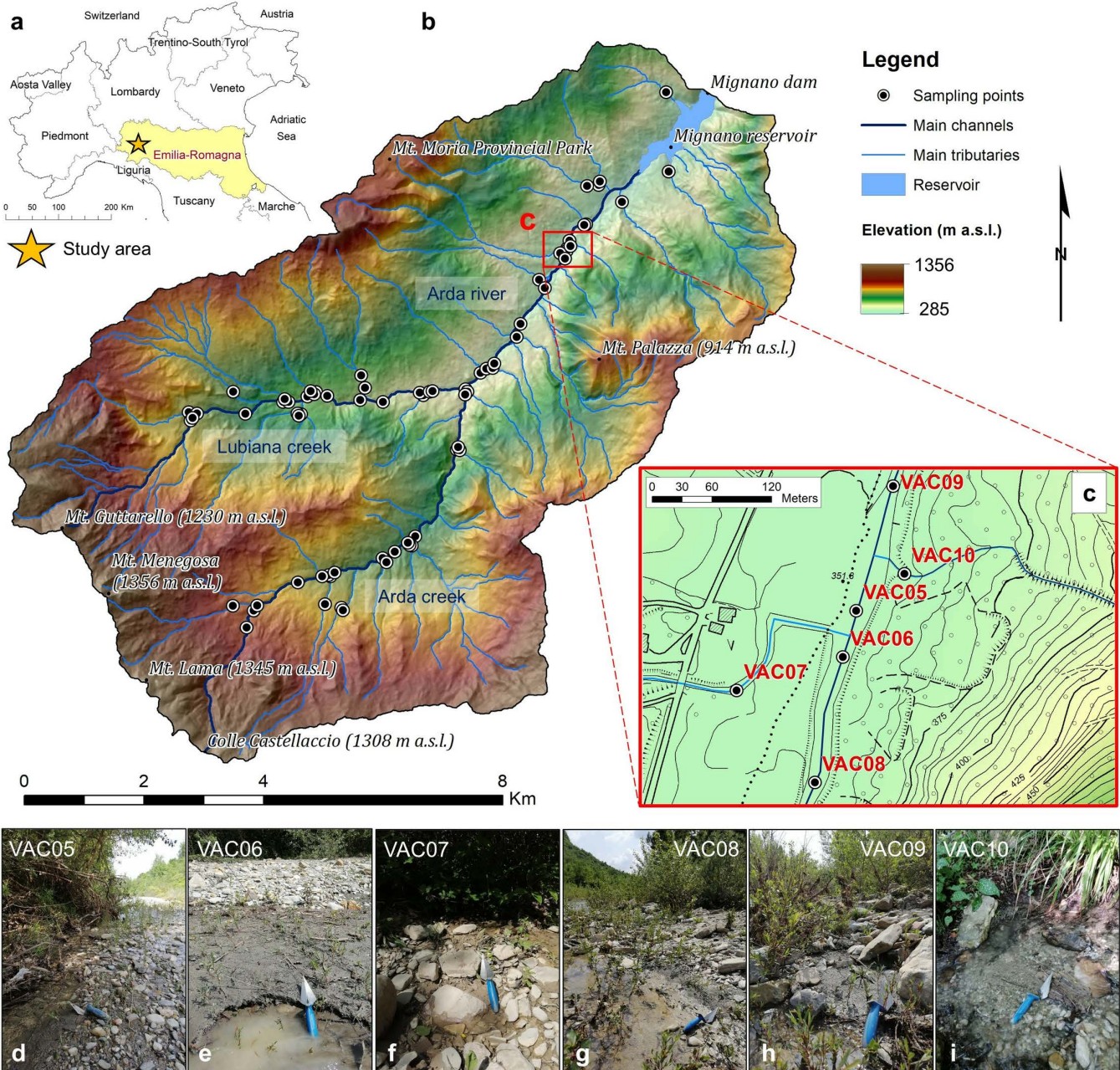

**Fig 1. (a) Geographic outline of the study area in the Emilia-Romagna Region, Northern Italy.** (b) The upper Val d'Arda-Mignano watershed. The main elevation peaks, elevation distribution, hydrographic elements, and other relevant toponyms are represented on the map. The location and distribution of sampling sites (fluvial sediments) are also reported. Base Map: Digital Terrain Model 5x5, Ed. 2014 (© Archivio Cartografico, Regione Emilia-Romagna; https://metasfera.regione.emilia-romagna.it/ricerca_metadato?uuid=r_emiro:2016-08-08T155835) (c) Explanatory excerpt showing the location of some sampling sites along the main channels and stream tributaries (d, e, f, g, h, i). Base Map: Technical Regional Map 1:5,000 (© Archivio Cartografico, Regione Emilia-Romagna; https://metasfera.regione.emilia-romagna.it/ricerca_metadato?uuid=r_emiro:2017-11-24T113154).

and covers around 88 km², with elevation ranging from 285 to 1356 m a.s.l (Fig 1b). In the upper part of the study area, the watershed is drained by two main channels, which converge into the Arda river before flowing into the Mignano reservoir: the Lubiana and Arda creeks (Fig 1b).

The study area represents a typical Mediterranean temperate watershed, characterized by decreasing mean annual temperature and increasing annual precipitation with altitude, and rainfall peaks in April, October, and November exceeding 100 mm. Summers are hot with minimal water deficit [47]. As with many Mediterranean watersheds, the study area consists in a complex palimpsest landscape, with features inherited from I) the past morphoclimatic evolution, II) the activities of past cultures and the remnants of geomorphic processes that partially determine the management of the land, and III) the actual geomorphic setting [9,58]. At the same time, it exhibits highly diversified environmental conditions, with substantial variability in lithological, geomorphological, and Land Use/Land Cover (LULC) characteristics, making it particularly suitable for geospatial watershed-scale modelling. Furthermore, its accessibility as an anthropized watershed provides good opportunities for intensive sampling campaigns and field-based research.

Particularly, the study area features geological formations dating from the Upper Cretaceous to the Paleocene-Eocene periods [59]. According to Martini & Zanzucchi [60], these formations can be grouped based on similar lithological characteristics: I) Silicified Calcilutites and Silty Clays, II) Varicoloured Clays and Shales, III) Carbonate Turbidites, IV) Ophiolitic/Sedimentary Breccias and Olistoliths, V) Arenaceous-Pelitic Turbidites. Landscape evolution is shaped by the interaction of lithological characteristics and local structural features [61]. The watershed alternates morphological traits typical of hilly and mountainous settings, thereby influencing both geomorphic processes and vegetation patterns [47].

The geological setting significantly drives the complex geomorphology of the study area, along with peculiar morpho-structural characteristics. The prevalence of 'weak rocks' with a prevalent clayey component makes the watershed highly prone to intense erosion and large mass movements [47,62]. Landslides are the dominant land degradation process, exhibiting considerable variability in magnitude and frequency. In the study area, landslide activity is largely dominated by periodic reactivation of pre-existing large-scale landslide bodies, primarily triggered by intense or prolonged rainfall events. Fluvial undercutting at the toe of landslide deposits also plays a critical role in destabilizing valley slopes. Also, streambank and upland water erosion contribute to severe land degradation, facilitating sediment delivery to the channel network and depositing substantial sediment volumes in the Mignano reservoir [63]. Erosion processes like gullying and subsurface erosion are constrained by lithological features, particularly associated with claystones.

Forest vegetation dominated by oaks, hornbeams, and chestnuts is the most representative LULC type in the watershed, shaping the hilly landscape at lower elevations, while beech forests prevail in mountainous areas at higher altitudes. Water erosion processes such as rill-interrill erosion and gullying appear to be less prevalent in forested environments. This is likely due to a combination of the protective effect of the canopy against rainfall and the inherent difficulty in identifying lower-magnitude erosion processes, especially considering that these areas are generally less accessible for detailed field investigations. Coniferous coppice woods are also common, especially because they are employed to restore degraded and eroded steep slopes. Rainfed arable lands represent the predominant agricultural landscape in the watershed but are significantly affected by rill-interrill erosion, which is strongly influenced by seasonal variations due to changing land cover [64].

The soils of the study area are affected by biochemical alteration and decarbonation, with *Calcaric Cambisols* being dominant on various parent materials. *Calcaric Regosols* are found on steep, erosive slopes with scarce vegetation, while *Eutric Cambisols* and *Dystric Cambisols* are present on more stable slopes [65,66].

## 3 Materials and methods

In order to accomplish the objectives of this study, we adopted an interdisciplinary methodological workflow as shown in Fig 2: 1) to assess and visualize geospatial data and prepare the sampling design, we delineated the watershed boundaries and the channel network through GIS-based hydrological modelling from a Digital Terrain Model (DTM); 2) we

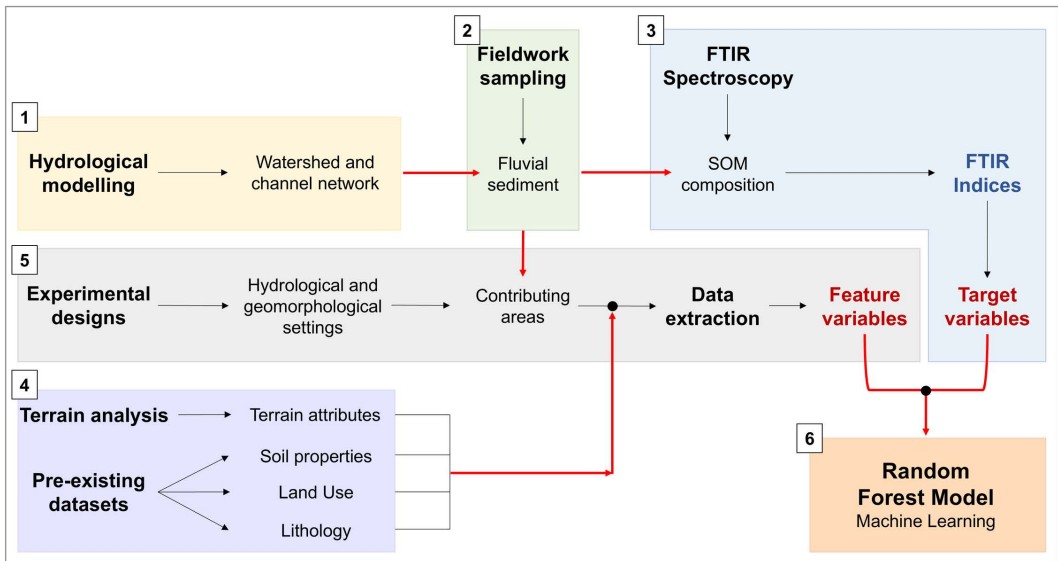

**Fig 2. Schematic methodological workflow used in the present study.**

conducted a sampling campaign to collect fluvial sediments across the watershed from the main channels and their tributaries; 3) samples collected were analysed through Fourier Transform Infrared (FTIR) spectroscopy to characterize SOM composition and calculate four different FTIR proxies (i.e., indices) as potential indicators of soil erosion susceptibility. A dataset of 'target variables' was then compiled using values of the calculated indices; 4) several environmental variables including terrain features, soil properties, lithological and LULC datasets were collected in a GIS environment; 5) for each sampling point, we extracted data of the environmental variables from the respective upstream contributing areas through three experimental designs, accounting for different hydrological and geomorphological settings. Then, three datasets of 'feature variables' were compiled using values extracted from the variables used; 6) the datasets were finally analysed through a RF machine learning approach to assess the non-linear and geospatial relationships between target and feature variables.

Access to the study area and sampling sites did not require any permission from local authorities, as the watershed is an open-access area, and no private property was violated in any way.

In the present study we used the Projected Coordinate System WGS84 – UTM 32N (EPSG: 32632).

## 3.1 Watershed and channel network delineation

The selected watershed was delineated using the System for Automated Geoscientific Analyses (SAGA) [version 8.1.1; [67] based on a 5 m DTM of the Emilia-Romagna Region [RER; 68]. A Gaussian Filter of radius 3 was applied for filtering errors and artifacts. The DTM was pre-processed for hydrological modelling using the *Deepen Drainage Routes* tool, with a threshold of 10 m [69]. Then, we derived channel network and watershed boundary by computing Fow Directions [70] and Flow Accumulation [71,72] (Fig 2). A Strahler order of 5 was used as threshold for the computation of the main channel network [73].

## 3.2 Sampling of fluvial sediments

To accomplish a representative characterization of soil and associated organic matter eroded from hillslopes in the upper Val d'Arda, we collected a total of 73 fluvial sediment samples (i.e., VAC 01–73; Fig 1b; Fig 2). We systematically collected

samples along the main channels (i.e., Lubiana creek and Arda river) and main tributaries, particularly before and after their intersections (Fig 1c). The sampling design was aimed at capturing a representative cross-section of channel bed sediments, assumed to reflect the material transported from upstream contributing areas to the respective downstream sampling points [74; i.e., each sampling site is considered as the outlet of its upstream area]. Hence, our approach ensured comprehensive coverage of the watershed contributing areas.

The sampling campaign was carried out at the end of May 2023, right after an intense hydro-meteorological event that impacted the Emilia-Romagna Region [75,76]. Hence, freshly deposited river sediments were collected shortly after the heavy rainfall event. These sediments were identified by several characteristics indicative of recent deposition, including: I) a light greyish colour; II) a very poorly weathered appearance, unlike older, oxidized deposits; III) a well-sorted grain size distribution ranging from clay to coarse sand; IV) the presence of thin and horizontal layering as well as cross-bedding, ripple marks and micro-dunes; and V) the absence of vegetation or significant biological activity above the surface (e.g., Fig 1d-i). This uniformity in deposition conditions among the samples is critical for maintaining high organic content and preserving biochemical properties [77]. Moreover, samples were collected at locations associated with reduced flow velocity, including small point bars, riffle-pool sequences, and areas downstream of physical obstacles. When sediments are deposited under similar hydrodynamic conditions, they generally exhibit more consistent physical and biochemical properties, which is crucial for comparative analysis [78,79]. Based on these considerations, we assume that the differences observed in the sampled material (Fig 1) can be attributed to a set of environmental variables that influenced the chemical composition of SOM from its origin in the upstream areas to its deposition at the sampling sites.

Sampling along tributaries was carried out at locations close to the intersection with the main stream. In the latter case, particular attention was paid to select locations that remain unaffected by water rise from the main stream during flood events, thus maintaining the integrity of the samples. Four replicates were collected at each sampling site cross-section and later mixed in the field to obtain a representative composite sample [80]. The samples were then air-dried, disaggregated, homogenized, and sieved to ≤ 2 mm [81].

### 3.3 Fourier transform infrared (FTIR) spectroscopy analysis

We analysed SOM functional groups for the 73 fluvial sediment samples through FTIR spectroscopy, using the potassium bromide (KBr) transmission technique, following Ellerbrock et al. [82] and Ellerbrock & Gerke [50,83] (Fig 2). The laboratory procedure, along with equipment used, software, and specific settings, is reported in S1 Appendix (Supporting Information). In the end, we obtained the absorption spectra of sediment samples in a range of wavenumbers (WN) between 3900 and 400 cm$^{-1}$. We further processed and interpreted spectra data using a common procedure, as reported in Ellerbrock et al. [84].

The FTIR spectra exhibit bands that represent absorption of infrared light at frequencies (i.e., wavenumbers, WN; cm$^{-1}$) specific to the type and excitation behaviour of the sample [49]. Here, we focused on specific WN regions of the spectra to characterize the following functional groups: I) C–H (band A), II) C=O (band B), III) Carbonate/$v_s$COO$^-$ (band C), IV) C-O-C/Si-O-Si (band D) (Fig 3). The C–H functional groups (i.e., hydrophobic alkyl groups) were analysed in the WN region 3020−2800 cm$^{-1}$ [85] (Fig 3). Within this range, the bands at ~ 2920 cm$^{-1}$ and ~ 2860 cm$^{-1}$ are related to asymmetric ($A_1$) and symmetric ($A_2$) stretching vibrations of methyl and methylene groups. The absorption intensities of bands $A_1$ and $A_2$ were measured as the vertical distance (i.e., height) from a 'baseline' plotted between tangential points on absorption minima to the $A_1$ and $A_2$ maxima, following Ellerbrock et al. [86]. Then, the $A_1$ and $A_2$ bands were combined into a single band A (Fig 3). The C=O functional groups typically cause bands in the following WN regions: I) 1740−1698 cm$^{-1}$ (typical for carboxyl groups in ketones, carboxylic acids, or amides) and II) 1640–1600 cm$^{-1}$ (asymmetric carboxylate stretching, i.e., $v_{as}$COO$^-$ band) [87] (Fig 3). The C=O band intensities were measured as the height from the total baseline of the spectra to the respective band maxima [84]. Hence, the band intensities within the before mentioned WN regions are denoted as Bands $B_1$ and $B_2$, respectively. Then, the $B_1$ and $B_2$ bands were combined into a single band B (Fig 3).

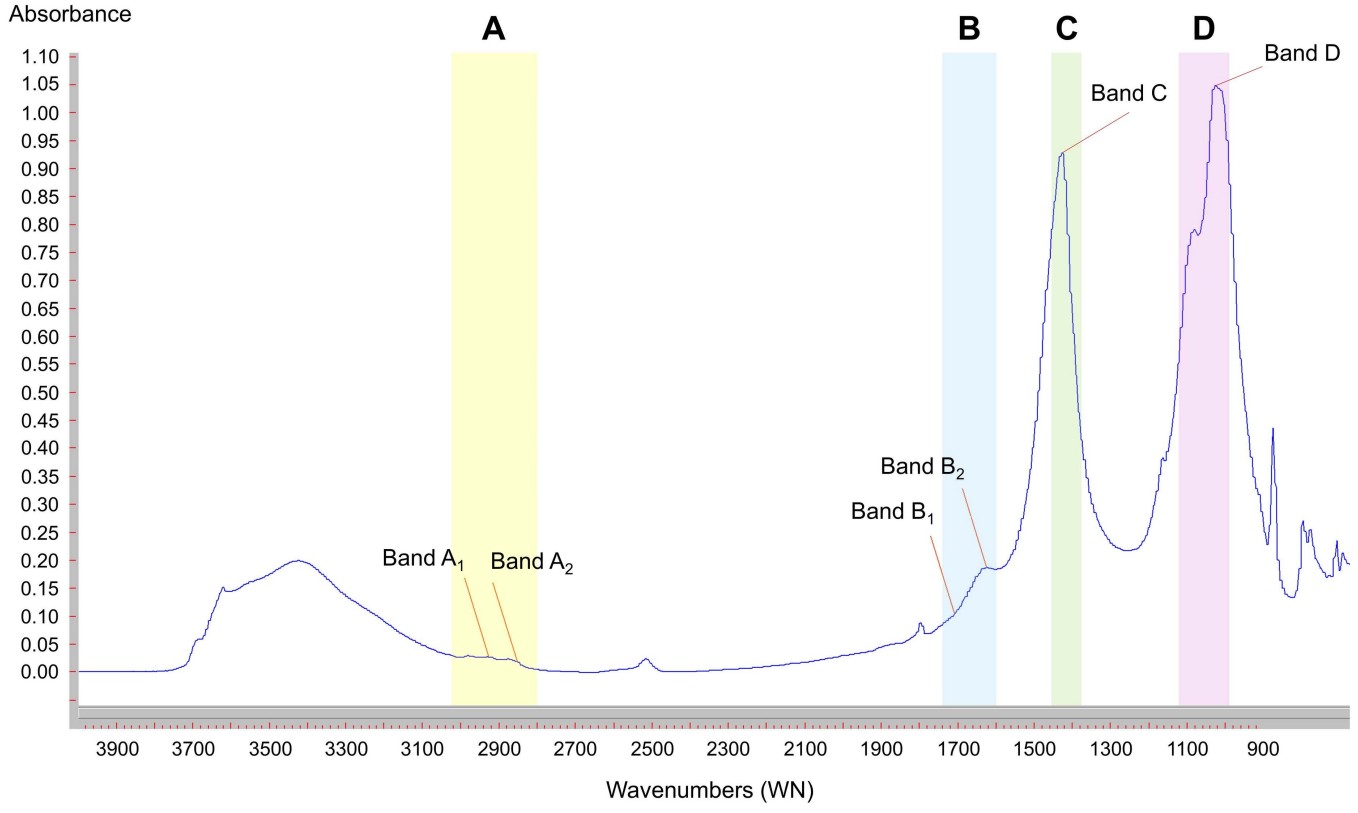

**Fig 3. Identification of the bands A, B, C, and D, within specific wavenumber regions, on a smoothed, baseline-corrected spectrum.**

Likewise, the C=O groups cause a third band in the WN region 1450−1382 cm⁻¹, which accounts for the symmetric vibration of the carboxylate groups [$\nu_s$COO⁻ band; 88,89]. However, in FTIR analysis of carbonate-rich soils the $\nu_s$COO⁻ band is strongly overlapped by the carbonate band [90], which shows a maximum intensity at around 1420 cm⁻¹ [53]. Therefore, according to Alvarez-Puebla et al. [54], the $\nu_s$COO⁻/carbonate band (i.e., band C; Fig 3) intensity may be related to the formation of organic matter-cation associations due to a potentially high Ca²⁺ cation content. Finally, the C-O-C functional groups, indicating the presence of polysaccharides or ether groups often associated with hydrophilic components of SOM [84], were analysed in the WN region 1120−1000 cm⁻¹ [55; band D] (Fig 3). However, in soil/sediment samples, the band D may be overlapped by an intense Si-O-Si band due to the presence of silicates and aluminosilicates.

We calculated, for each sample, four indices as the ratios of the band peaks (i.e., A/B, A/D, B/D, and C/D; Table 1; Fig 2). Then, a dataset of 'target variables' was compiled using values of the calculated indices (Fig 2). Afterwards, a Spearman's rank correlation analysis was performed to assess the rank correlation between the indices. Moreover, the index values were classified into 10 classes according to the Jenks natural breaks method [91] and displayed in a GIS environment to identify and visualize spatial patterns.

### 3.4 Preparation of environmental variable datasets

In this study, we used several watershed-scale raster datasets as environmental variables, including terrain (morphometric) features and soil properties, as well as vector data on lithology and LULC (Fig 2).

In particular, we performed a DTM-based terrain analysis using the SAGA GIS software to characterize the main land-surface features which potentially influence hydrological processes, soil erosion and, in general, sediment dynamics

**Table 1. Indices for SOM characterization calculated as FTIR band ratios, used in the present study.**

| Index | Bands ratio | Meaning | References |
|---|---|---|---|
| $\frac{A}{B}$ | $\frac{(A_1+A_2)}{(B_1+B_2)}$ | Relative hydrophobicity of SOM. A higher ratio indicates a higher content of hydrophobic groups, suggesting that the sediment has a reduced wettability. | [51] [52] |
| $\frac{A}{D}$ | $\frac{(A_1+A_2)}{D}$ | Relative hydrophobicity of SOM. A higher ratio indicates a higher content of hydrophobic groups, suggesting that the sediment has a reduced wettability. | [84] |
| $\frac{B}{D}$ | $\frac{(B_1+B_2)}{D}$ | Relative Cation Exchange Capacity (CEC) of SOM. A higher ratio is often correlated with a higher CEC because carbonyl and carboxyl groups can participate in cation exchange processes. | [39] [50] |
| $\frac{C}{D}$ | $\frac{C}{D}$ | For SOM, it can be used as a relative measure of organic matter-cation associations. A higher ratio suggests a greater degree of interaction between SOM and cations. | [54] [55] |
| | | Depending on the mineral composition, it is a measure of the relative carbonate content in the sample. | [53] |

[92,93] (Fig 2). Table 2 summaries the terrain variables used in this study, along with a description of their geomorphological significance. As they were calculated from the DTM [68], the final resolution of the outputs is 5 m. Detailed information on the methods used for calculating these variables is included in S1 Appendix (Supporting Information).

Moreover, the RER provided the regional raster datasets of soil physical and chemical properties (https://datacatalog.regione.emilia-romagna.it/catalogCTA/) (Fig 2). The datasets, which are related to the topsoil (0–30 cm), were elaborated at 1:50,000 scale by means of a Digital Soil Mapping approach, with a pixel resolution of 100 m. In the present study, we considered the following soil properties:

- Soil Organic Carbon (SOC) stock (expressed in Mg ha$^{-1}$) [106];

- Soil Organic Carbon (SOC) content (expressed in %) [107];

- Clay content (particles with diameter ≤ 2 μm, in %) [108];

- Silt content (particles with diameter > 2 μm and ≤ 50 μm, in %) [108];

- Sand content (particles with diameter > 50 μm and ≤ 2 mm, in %) [108];

- Skeleton content (particles with diameter > 2 mm, in %) [108];

- Soil pH [109].

All the datasets are provided along with the related pixel-based cartographic accuracy (i.e., low, medium, or high accuracy), defined as the standard deviation of the estimated value in each pixel (i.e., the lower the standard deviation, the more accurate the estimate). For the study area, all the datasets have, on average, a medium accuracy. Further details concerning the full methodology (i.e., spatial prediction, validation, regionalization, and accuracy evaluation) can be found in Ungaro et al. [106–109]. Afterwards, all the datasets were resampled at 5 m resolution using the Nearest Neighbour method. Due to the nature of the data, the initial resolution and processing methods of these datasets may potentially introduce errors and uncertainties related to data representation and spatialization. Nevertheless, resampling and alignment with outputs derived from the DTM ensures consistency in spatial resolution, which is essential for accurate and homogeneous pixel value sampling across all datasets and for reliable statistical analysis.

Additionally, the RER also provided the regional shapefile datasets of land use and lithological data (Fig 2). In particular, the following datasets were used:

- Geological units, 1:10,000 scale [110];

- Land use, 1:10,000 scale [111].

**Table 2. Terrain variables used in the present study, along with their geomorphological significance. Sources: [92,94–102].**

| Terrain variable | Method/ Reference | Unit | Geomorphological significance |
|---|---|---|---|
| Elevation (DTM) | [68] | Meters | Climate, vegetation, potential energy. |
| Flow Accumulation | [72] | Meters$^2$ | Contributing catchment area, steady-state runoff rate. |
| Slope | [103] | Degrees | Precipitation, overland and subsurface flow velocity and runoff rate, soil water content. |
| Aspect | [103] | Degrees | Flow direction, solar insolation, evapotranspiration, influence on vegetation distribution. |
| Profile curvature | [103] | 1/ meters | Flow acceleration and deceleration, soil erosion and deposition rates. |
| Tangential curvature | [103] | 1/ meters | Local flow convergence and divergence. |
| General curvature | [103] | 1/ meters | General measure of terrain convexity, identifying convex surface as interfluves and peaks and concave surface as valleys and cavities. |
| Total curvature | [103] | 1/ meters | Comprehensive measure that sums up all components of surface curvature, influencing water flow and erosion patterns. |
| Downslope Distance Gradient | [104] | Meters | Rate of elevation change over a horizontal distance, providing a measure of slope steepness and flow direction for understanding flow paths, erosion potential, and landscape stability. |
| SAGA Wetness Index | [92] | Dimensionless | Spatial distributions and extent of areas of high soil moisture and saturation excess overland flow as a function of upslope contributing area, soil transmissivity, and slope. |
| Terrain Ruggedness Index | [105] | Meters | Heterogeneity or ruggedness of the terrain (terrain complexity). |
| Stream Power Index | [94] | Meters | Potential erosive power of flowing water (based on the assumption that discharge is proportional to the specific catchment area). |
| Channel Network Base Level | [102] | Meters | Erosion potential and sediment transport/deposition efficiency in relation to the landscape topography. |
| Vertical Distance to Channel Network | [102] | Meters | Identification of fluvial terrace systems, palaeo-surfaces, planation surfaces, and erosion and deposition zones. |

Data were managed using the Esri© ArcMap software (version 10.3.1; ArcGIS). Then, geological units were merged based on lithological characteristics according to Servizio Geologico d'Italia [59] and Martini & Zanzucchi [60], obtaining 5 lithological groups as shown in Table 3. Also, land use classes were grouped to obtain 10 LULC types as shown in Table 4 (rivers and water bodies were excluded).

### 3.5 Variables extraction based on different experimental designs

We extracted data related to the variables reported in the previous section from the contributing areas (CAs) of the sampling points, in a GIS environment (Fig 2). In this study, we consider the CA of each sample as the region where soil erosion and downstream sediment transport occur, with the sampling site representing the respective outlet. To determine the best strategy for extracting data from CAs, we set up three different experimental designs, representative for different hydrological and geomorphological settings (Fig 4):

1. **Experiment 1**. We used the entire hydrological areas upstream of each sampling point (hereinafter called 'Upstream areas Exp_1'; Fig 4a). In particular, CAs are calculated with the Deterministic 8 method following O'Callaghan & Mark [112].

2. **Experiment 2**. CAs were narrowed to sediment sources and sinks (hereinafter called 'Sources and sinks Exp_2'; Fig 4b), without discriminating between the type of landform/process, the activity status, the connectivity to the drainage system, nor between the erosional or depositional dynamic. We used the entire shapefile mask (i.e., "dissolved") of the 'Inventory Map' (IM) of sediment sources and sinks of the upper Val d'Arda published in La Licata et al. [62,113]. The latter was subsequently clipped within single CAs.

**Table 3. Lithological groups of the upper Val d'Arda-Mignano watershed. Shapefile data derived from the Geological Units database of the Emilia-Romagna Region [1:10,000 scale; 110]. The description of the lithological groups was adapted from the Geological Sheet 198 – Bardi [1:50,000 scale; 59].**

| Label | Lithological group | Description |
|---|---|---|
| LI_1 | Silicified Calcilutites and Silty Clays | Silicified calcilutites in medium to thick beds with intercalations of marly limestones. Silty clays with frequent intercalations of thin-bedded fine turbiditic sandstones (e.g., Palombini Shales). |
| LI_2 | Varicoloured Clays and Shales | Varicoloured clays and shales with intercalations of fine turbiditic sandstones and thin layers of calcilutites (e.g., Cassio Varicoloured Clays). |
| LI_3 | Carbonate Turbidites | Turbidites composed of calcareous marls, marly limestones, and marls with an arenitic or calcarenitic base. Alternation of sandstones, shales, calcareous marls, and marls (e.g., Monte Cassio Flysch). |
| LI_4 | Ophiolitic/Sedimentary Breccias and Olistoliths | Matrix-supported and grain-supported breccias, containing sedimentary and ophiolitic elements. Olistoliths made up of calcilutites, sandstones, serpentinites, basalts, granites, and jaspers (e.g., Ophiolitic Sandstones of the Pietra Parcellara Complex). |
| LI_5 | Arenaceous-Pelitic Turbidites | Alternations of arenaceous-pelitic rocks in thin to medium beds, with rare intercalations of marls and thin to medium bedded laminated limestones (e.g., Scabiazza Sandstones). |

**Table 4. LULC types of the upper Val d'Arda-Mignano watershed. Shapefile data derived from Land use database of the Emilia Romagna Region [1:10,000 scale; 111].**

| Label | LULC type |
|---|---|
| LU_1 | Coniferous forests |
| LU_2 | Beech forests |
| LU_3 | Oak, hornbeam and chestnut forests |
| LU_4 | Riparian vegetation |
| LU_5 | Cultivated fields |
| LU_6 | Anthropic areas |
| LU_7 | Bare surfaces/ rocky outcrops |
| LU_8 | Sparse vegetation in evolution |
| LU_9 | Meadows |
| LU_10 | Bushes and shrubs |

3. **Experiment 3**. CAs were further narrowed to the geomorphological areas with the potential to produce sediment and deliver it to the main channels (hereinafter called 'Hazard areas Exp_3'; Fig 4c). We employed the 'Hazard Map of sediment Production and Delivery' (HPD) of the upper Val d'Arda published in La Licata et al. [62], only focusing on the areas characterized by a medium, high, or very high hazard for sediment sourcing and delivery. These hazard-prone areas mostly include landslides highly connected to the drainage system and overlaid by additional processes like rill-interrill erosion, gullying, stream incision, and bank erosion, as well as steep rock walls affected by rockfalls and debris flows. The HPD was generated as output of the HOTSED model, covering geospatial information of sediment sources/sinks and related dynamics with the assessment of structural and functional properties of sediment connectivity [62,114]. Also in this case, the HPD map was clipped within single CAs.

Finally, we extracted for each experiment data of the environmental variables, as follows:

- **Mean value of the raster in the CA**: *Elevation*, *Slope*, *Aspect*, *Profile Curvature*, *Tangential Curvature*, *General Curvature*, *Total Curvature*, *SAGA Wetness Index*, *Terrain Ruggedness Index*, *Stream Power Index*, *Channel Network Base Level*, *Vertical Distance to Channel Network*, *SOC stock*, *SOC %*, *Clay %*, *Silt %*, *Sand %*, *Skeleton %*, and *pH*.

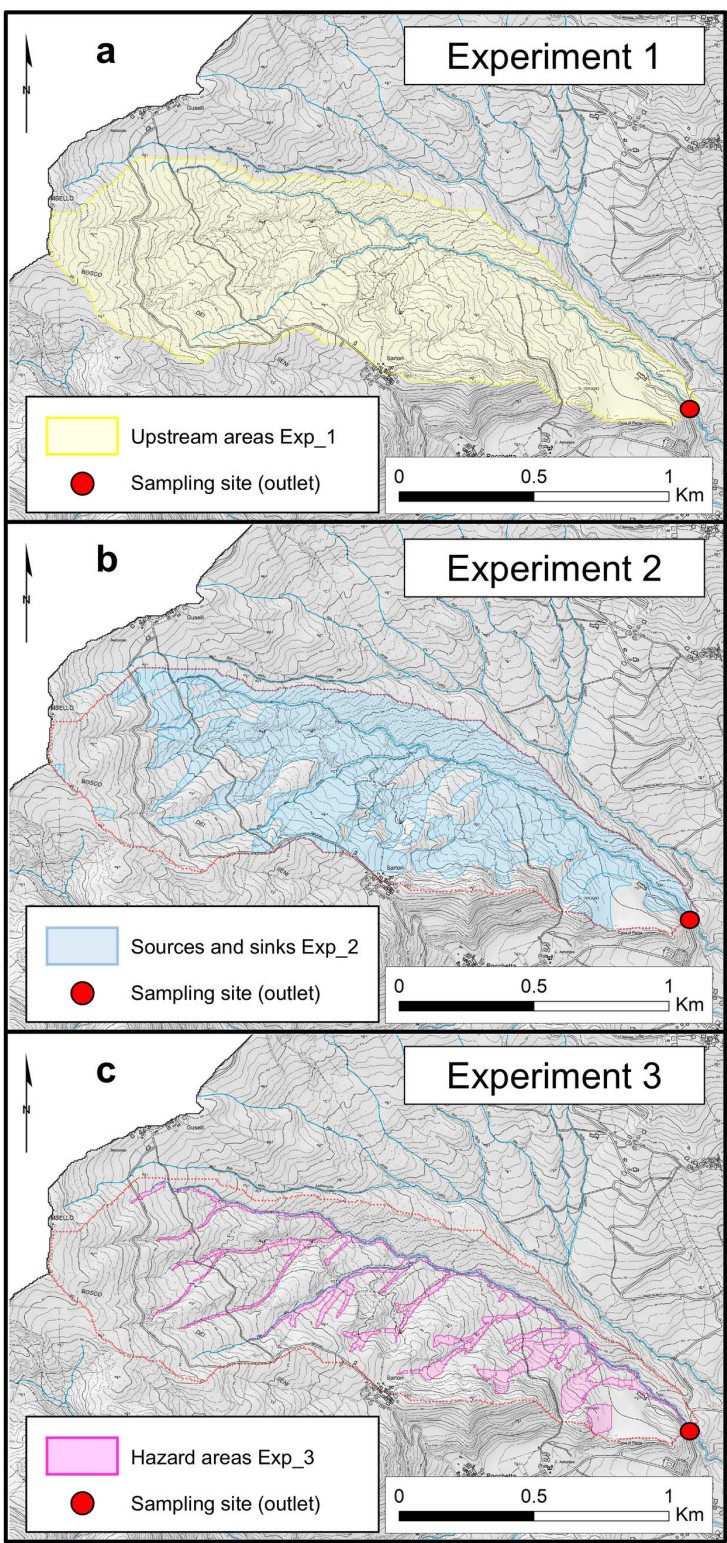

**Fig 4. The three experimental designs employed in the present study: (a) 'Upstream areas Exp_1', (b) 'Sources and sinks Exp_2', (c) 'Hazard areas Exp_3'.** The coloured areas delimited in the three experimental designs were used to extract values from the feature variables. This methodology was replicated for each of the 73 sediment samples (Fig 1b). Base Maps: Hillshade Map derived from DTM 5x5, Ed. 2014 (© Archivio Cartografico,

Regione Emilia-Romagna; https://metasfera.regione.emilia-romagna.it/ricerca_metadato?uuid=r_emiro:2016-08-08T155835) and Technical Regional Map 1:5,000 (© Archivio Cartografico, Regione Emilia-Romagna; https://metasfera.regione.emilia-romagna.it/ricerca_metadato?uuid=r_emiro:2017-11-24T113154). Maps produced using the Esri© ArcMap software (version 10.3.1; ArcGIS).

- **Absolute value of the raster at the sampling point (outlet of the CA)**: *Flow Accumulation* and *Downslope Distance Gradient.*

- **Proportion of each class in the CA for lithological groups and LULC types**: *Silicified Calcilutites and Silty Clays*, *Varicoloured Clays and Shales*, *Carbonate Turbidites*, *Ophiolitic/Sedimentary Breccias and Olistoliths*, *Arenaceous-Pelitic Turbidites*, *Coniferous forests*, *Beech forests*, *Oak, hornbeam and chestnut forests*, *Riparian vegetation*, *Culti-vated fields*, *Anthropic areas*, *Bare surface/Rocky outcrops*, *Sparse vegetation in evolution*, *Meadows*, and *Bushes and Shrubs.*

In addition, we calculated the *Altitude Difference* as the difference between the mean elevation of the CA and the elevation at the sampling point. Moreover, the *Area*, *Length*, and *Width* of each CA were calculated as geometric parameters. Specifically, *Length* and *Width* were determined as the longest and shortest sides of a minimum bounding rectangle that encloses the CA, respectively.

In the end, we obtained three different datasets related to the three experimental designs, each containing 40 'feature variables'.

### 3.6 Determination of non-linear and geospatial relationships using Random Forest algorithm

The datasets were analysed through a Random Forest (RF) model to determine the non-linear and geospatial relationships between target and feature variables (Fig 2). Initially, feature selection was performed using the Spearman's correlation filter from the 'FSelector' package in R [115]. We selected the top 10 most significant features for each target variable, helping to streamline the data and focus on the most influential variables. The RF algorithm, which is an ensemble machine learning method [56], was then used to model the relationships between the selected features and target variables. RF operates by constructing multiple decision trees during training and outputting the mean prediction of the individual trees. It excels in handling complex and non-linear relationships by averaging the results from numerous trees to improve accuracy and control overfitting [57]. The RF prediction is performed using the *rf* function in the 'caret' package in R [116,117].

To evaluate model performance, we implemented Leave-One-Out Cross-Validation (LOOCV) on the entire dataset as it is beneficial for small datasets, detects overfitting, and provides error estimates with comparatively good bias and variance properties [118]. In LOOCV, each data point is used once as a test set while the remaining data points form the training set. This technique ensures that every observation is tested exactly once, providing a comprehensive evaluation of model performance. Within each training dataset, we performed a Grid Search to optimize the 'mtry' parameter of the RF model. The 'mtry' parameter, which controls the number of variables randomly sampled as candidates at each split in the decision trees, was tuned between 2 and 10 using 10-fold cross-validation. The optimal 'mtry' value was used to develop the final model, which was tested on the single data point left out in each iteration of LOOCV.

The performance of each RF model was assessed using R-square ($R^2$), Lin's Concordance Correlation Coefficient (CCC), Mean Squared Error (MSE), Root Mean Squared Error (RMSE), Bias, and Nash-Sutcliffe Efficiency (NSE), as commonly defined in literature [e.g., 57,119–121]. These metrics were performed with the help of *goof* function from the 'ithir' package [122] and 'hydroGOF' package in R [123].

For each model developed, post-hoc interpretability was conducted using the 'iml' package in R [124]. We first assessed the importance of each feature using the variable permutation method. This technique evaluates the effect of permuting (shuffling) each feature on the model's performance, thereby indicating the importance of each feature in

predicting the target variable. Additionally, we employed Accumulated Local Effects (ALE) plots to explore interactions between features and target variables. ALE plots provide insights into how the predicted values change as a feature varies, accounting for interactions with other features. The ALE plot helps in understanding the local effect of each feature while controlling for the effects of other features, thus revealing nuanced relationships and interactions [125].

## 4 Results

### 4.1 FTIR indices for SOM characterization (target variables)

The FTIR analysis carried out on collected samples (i.e., VAC 01–73; Fig 1b) provided the intensity of the band maxima related to selected SOM functional groups (Fig 3): I) hydrophobic alkyl groups (Bands $A_1$ and $A_2$), II) hydrophilic carboxyl and carboxylate groups (Bands $B_1$ and $B_2$), III) carbonate content/symmetric vibration of $COO^-$ (Band C), and IV) polysaccharides hydrophilic groups (C-O-C; Band D) (Fig 3).

Over the 73 analysed samples, Band $A_1$ has a mean value of 0.005, a median of 0.004, and a standard deviation (STD) of 0.003. Band $A_2$ has a mean value of 0.01, a median of 0.01, and a STD of 0.04. Band $B_1$ has a mean value of 0.11, a median of 0.11, and a STD of 0.03. Band $B_2$ has a mean value of 0.20, a median of 0.19, a STD of 0.05. Band C has a mean value of 0.86, a median of 0.77, and a STD of 0.51. Band D has a mean value of 1.32, a median of 1.24, and a STD of 0.42.

Furthermore, 4 indices were calculated as FTIR band ratios (Table 1) to characterize the SOM in the samples, in terms of relative hydrophobicity (i.e., A/B and A/D), relative CEC (i.e., B/D), and relative intensity of organic matter-cation interactions or relative carbonate content (i.e., C/D) (Table 5). A/B and A/D show relatively low values with little variation, indicating that these indices are consistent across the samples. In particular, A/B index has a mean value of 0.04, a median of 0.04, and a STD of 0.02. Similarly, A/D index has a mean value of 0.01, a median of 0.01, and a STD of 0.01. Instead, B/D index shows more variability, but its values still remain within a moderate range. B/D index has a mean value of 0.25, a median of 0.24, and a STD of 0.09. Differently, C/D index exhibits the most significant variation, with a wide range indicating that the relative carbonate content, and consequently the interaction between SOM and calcium, varies considerably among the samples. C/D index has a mean value of 0.71, a median of 0.56, and a STD of 0.47.

Table 6 summaries the Spearman correlation coefficients among the calculated indices. Indices A/B and A/D are strongly correlated (i.e., 0.92), indicating that SOM hydrophobicity is well characterized in this study. The high correlation between A/D and C/D indices (i.e., 0.86) implies that hydrophobicity and relative carbonate content in the samples may be closely linked. However, even though the high correlation between B/D and C/D indices, their slightly lower coefficient value (i.e., 0.83) suggests that the two indices are capturing slightly different aspects of SOM-cation dynamics in the study area. The moderate correlation between A/B and C/D indices (i.e., 0.68), as well as the lower correlation between A/B and B/D indices (i.e., 0.55), suggests that the spatial distribution and accumulation of hydrophobic vs. hydrophilic SOM components might be driven by different landscape processes and dynamics (e.g., water fluxes, sediment erosion, transport, and deposition) and/or environmental conditions.

Considering the spatial distribution of collected samples (Fig 1b), A/B index varies significantly across the landscape, with higher values (i.e., indicating more hydrophobic SOM) concentrated in specific areas (Fig 5a). In the uppermost areas of the watershed, A/B values are generally lower, indicating a lower ratio of hydrophobic to hydrophilic functional groups (Fig 5a). In correspondence of the middle sections of the watershed there is a noticeable increase in A/B, whereas in the lower watershed, particularly near the confluence of main channels and larger tributaries, the index values are more variable. Main channels, particularly those flowing through the middle part of the watershed, tend to exhibit higher A/B values, while the tributaries generally show lower A/B values, especially in the upper and middle parts of the watershed (Fig 5a). Similarly, in the uppermost areas of the watershed, the A/D index is generally low (Fig 5b), suggesting a lower ratio of hydrophobic alkyl groups (C-H) to hydrophilic polysaccharides (C-O-C) and silicate (Si-O-Si) groups, while in the middle parts of the watershed the index values show some increase, although they remain relatively low overall (Fig 5b). In the

**Table 5. FTIR Indices calculated for the 73 fluvial sediment samples in the upper Val d'Arda-Mignano watershed (cfr. Table 1).**

| Sample | A/B | A/D | B/D | C/D | Sample | A/B | A/D | B/D | C/D |
|--------|-----|-----|-----|-----|--------|-----|-----|-----|-----|
| VAC01 | 0.067 | 0.021 | 0.306 | 0.994 | VAC38 | 0.028 | 0.006 | 0.215 | 0.377 |
| VAC02 | 0.075 | 0.052 | 0.689 | 2.012 | VAC39 | 0.010 | 0.002 | 0.176 | 0.275 |
| VAC03 | 0.026 | 0.006 | 0.220 | 0.601 | VAC40 | 0.021 | 0.003 | 0.155 | 0.334 |
| VAC04 | 0.056 | 0.017 | 0.296 | 0.708 | VAC41 | 0.034 | 0.008 | 0.244 | 0.996 |
| VAC05 | 0.039 | 0.013 | 0.322 | 1.428 | VAC42 | 0.043 | 0.010 | 0.239 | 1.070 |
| VAC06 | 0.060 | 0.013 | 0.224 | 0.764 | VAC43 | 0.037 | 0.009 | 0.244 | 0.695 |
| VAC07 | 0.074 | 0.034 | 0.465 | 1.655 | VAC44 | 0.044 | 0.013 | 0.285 | 1.136 |
| VAC08 | 0.037 | 0.012 | 0.325 | 1.435 | VAC45 | 0.032 | 0.008 | 0.236 | 0.722 |
| VAC09 | 0.054 | 0.014 | 0.257 | 1.222 | VAC46 | 0.045 | 0.012 | 0.272 | 0.724 |
| VAC10 | 0.059 | 0.021 | 0.360 | 0.433 | VAC47 | 0.046 | 0.007 | 0.162 | 0.331 |
| VAC11 | 0.044 | 0.015 | 0.335 | 0.830 | VAC48 | 0.011 | 0.002 | 0.202 | 0.175 |
| VAC12 | 0.037 | 0.012 | 0.321 | 1.005 | VAC49 | 0.016 | 0.003 | 0.179 | 0.214 |
| VAC13 | 0.067 | 0.023 | 0.348 | 1.878 | VAC50 | 0.041 | 0.010 | 0.237 | 0.488 |
| VAC14 | 0.040 | 0.009 | 0.234 | 0.764 | VAC51 | 0.014 | 0.003 | 0.183 | 0.295 |
| VAC15 | 0.031 | 0.007 | 0.234 | 0.532 | VAC52 | 0.027 | 0.006 | 0.219 | 0.495 |
| VAC16 | 0.025 | 0.006 | 0.253 | 0.384 | VAC53 | 0.021 | 0.005 | 0.237 | 0.502 |
| VAC17 | 0.047 | 0.006 | 0.125 | 0.174 | VAC54 | 0.033 | 0.007 | 0.204 | 0.584 |
| VAC18 | 0.016 | 0.004 | 0.235 | 0.267 | VAC55 | 0.030 | 0.011 | 0.367 | 0.822 |
| VAC19 | 0.005 | 0.001 | 0.219 | 0.233 | VAC56 | 0.031 | 0.008 | 0.277 | 0.910 |
| VAC20 | 0.012 | 0.002 | 0.134 | 0.166 | VAC57 | 0.037 | 0.012 | 0.318 | 1.215 |
| VAC21 | 0.045 | 0.009 | 0.209 | 0.377 | VAC58 | 0.025 | 0.005 | 0.184 | 0.389 |
| VAC22 | 0.054 | 0.011 | 0.206 | 0.542 | VAC59 | 0.042 | 0.012 | 0.300 | 1.179 |
| VAC23 | 0.039 | 0.007 | 0.174 | 0.559 | VAC60 | 0.015 | 0.003 | 0.189 | 0.310 |
| VAC24 | 0.012 | 0.002 | 0.131 | 0.176 | VAC61 | 0.047 | 0.008 | 0.166 | 0.271 |
| VAC25 | 0.027 | 0.006 | 0.208 | 0.550 | VAC62 | 0.033 | 0.005 | 0.147 | 0.116 |
| VAC26 | 0.029 | 0.004 | 0.139 | 0.206 | VAC63 | 0.019 | 0.004 | 0.219 | 0.269 |
| VAC27 | 0.011 | 0.002 | 0.165 | 0.201 | VAC64 | 0.052 | 0.013 | 0.241 | 1.107 |
| VAC28 | 0.013 | 0.002 | 0.148 | 0.195 | VAC65 | 0.036 | 0.012 | 0.321 | 1.400 |
| VAC29 | 0.013 | 0.002 | 0.175 | 0.219 | VAC66 | 0.046 | 0.013 | 0.272 | 0.788 |
| VAC30 | 0.056 | 0.016 | 0.276 | 0.888 | VAC67 | 0.020 | 0.005 | 0.225 | 0.446 |
| VAC31 | 0.033 | 0.007 | 0.208 | 0.480 | VAC68 | 0.040 | 0.007 | 0.182 | 0.537 |
| VAC32 | 0.052 | 0.016 | 0.312 | 1.286 | VAC69 | 0.049 | 0.011 | 0.213 | 0.531 |
| VAC33 | 0.051 | 0.019 | 0.369 | 2.070 | VAC70 | 0.040 | 0.010 | 0.242 | 0.641 |
| VAC34 | 0.017 | 0.004 | 0.241 | 0.385 | VAC71 | 0.041 | 0.014 | 0.336 | 0.871 |
| VAC35 | 0.048 | 0.016 | 0.345 | 0.846 | VAC72 | 0.093 | 0.022 | 0.241 | 1.341 |
| VAC36 | 0.049 | 0.012 | 0.246 | 0.961 | VAC73 | 0.067 | 0.017 | 0.251 | 1.300 |
| VAC37 | 0.024 | 0.005 | 0.222 | 0.342 | – | – | – | – | – |

lower watershed, near the confluence of main channels and tributaries, A/D values remain low to moderate, with fewer areas showing high ratios (Fig 5b). In general, both the main channels and tributaries display low A/D values, with only small increases in specific areas, suggesting minimal differences in hydrophobicity between these watercourse types. The B/D index exhibits a similar trend of A/B across the landscape, with a clear increase from the upper to the lower watershed (Fig 5c). The main channels show moderate to high B/D values (i.e., highlighting a greater proportion of carbonyl groups), particularly in the middle and lower watershed, while the tributaries generally exhibit lower values, especially in the upper

**Table 6. Spearman's rank correlation coefficients among the calculated FTIR indices (cfr. Table 1).**

|     | A/B | A/D | B/D | C/D |
| --- | --- | --- | --- | --- |
| **A/B** | 1.00 | 0.92 | 0.55 | 0.68 |
| **A/D** | 0.92 | 1.00 | 0.81 | 0.86 |
| **B/D** | 0.55 | 0.81 | 1.00 | 0.83 |
| **C/D** | 0.68 | 0.86 | 0.83 | 1.00 |

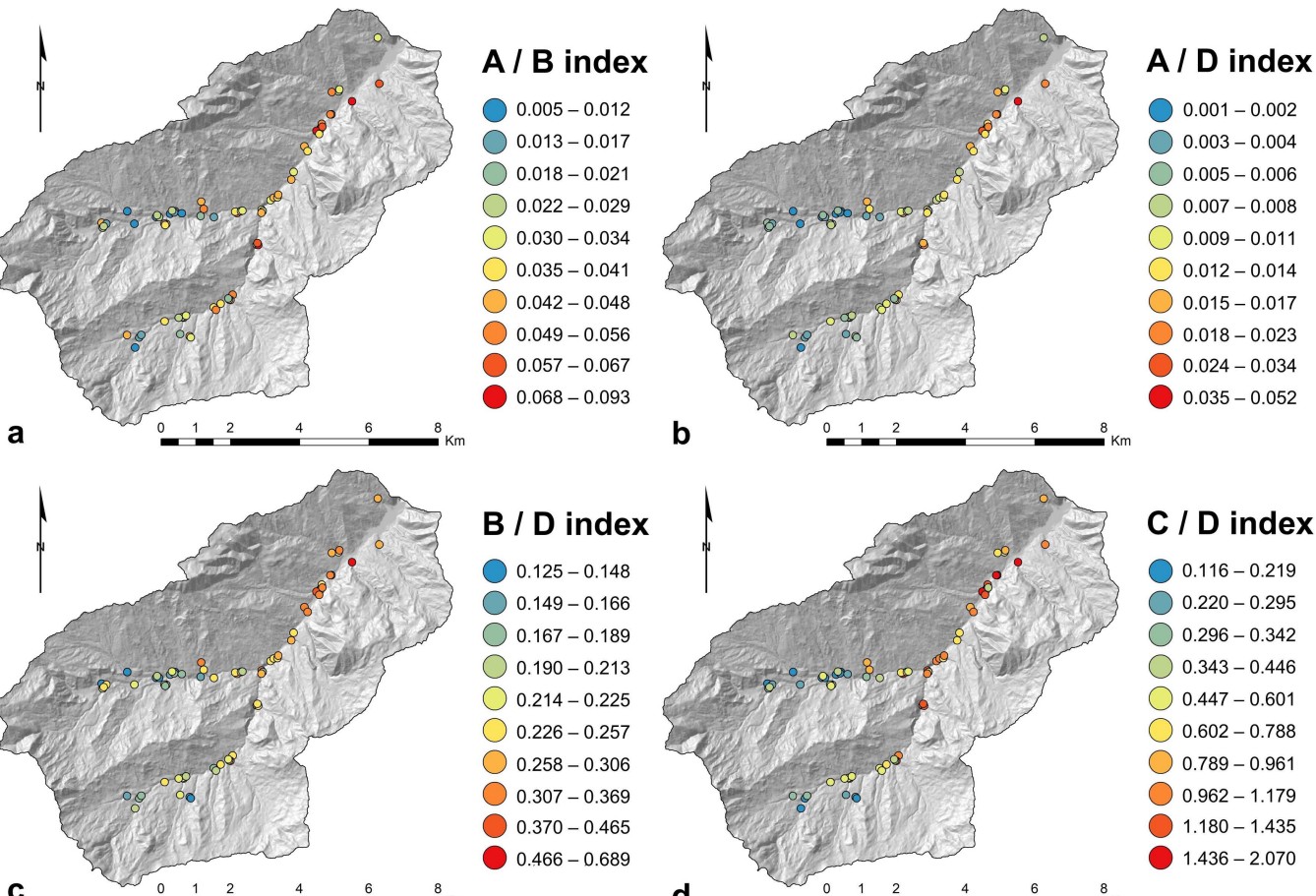

**Fig 5. Spatial distribution of FTIR indices (Table 1) within the upper Val d'Arda watershed (Fig 1b): (a) A/B, (b) A/D, (c) B/D, (d) C/D.** Values are classified into 10 classes according to the Jenks natural breaks method, for a better visual representation. Base Map: Hillshade Map derived from DTM 5x5, Ed. 2014 (© Archivio Cartografico, Regione Emilia-Romagna; https://metasfera.regione.emilia-romagna.it/ricerca_metadato?uuid=r_emiro:2016-08-08T155835). Maps produced using the Esri© ArcMap software (version 10.3.1; ArcGIS).

reaches (Fig 5c). Finally, the C/D index shows a noticeable increase from the upper to the lower watershed, particularly in the main channels (Fig 5d). Main channels have consistently higher C/D values compared to tributaries (Fig 5d).

## 4.2 Datasets selected as feature variables

The GIS-based terrain analysis provided 14 terrain variables that have been used as feature variables (Fig 6). Moreover, the raster datasets for 7 soil properties (i.e., SOC stock, SOC %, Clay %, Silt %, Sand %, Skeleton %, and pH) have been

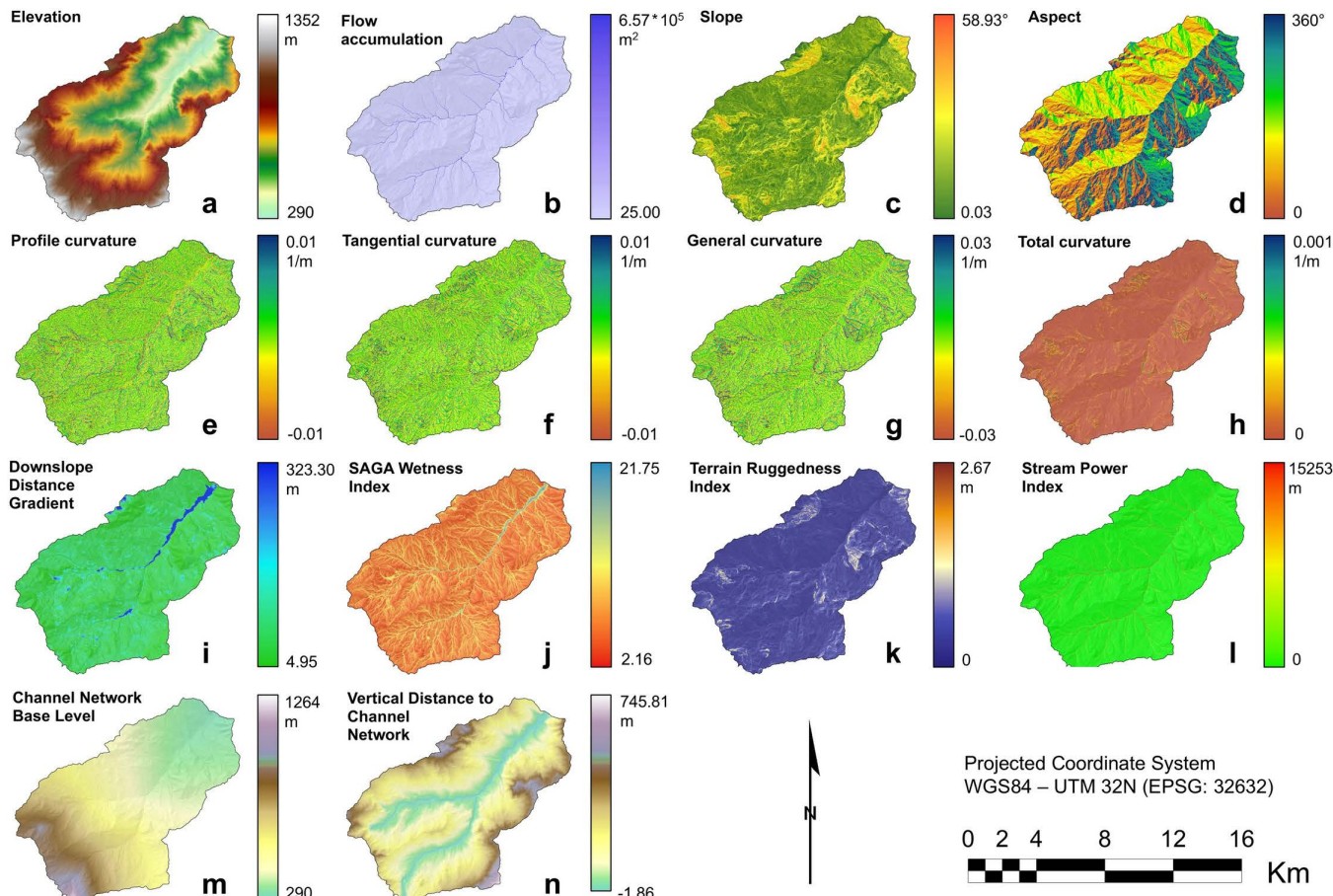

**Fig 6. Terrain variables of the upper Val d'Arda-Mignano watershed derived from GIS-based terrain analysis: (a) Elevation, (b) Flow Accumulation, (c) Slope, (d) Aspect, (e) Profile curvature, (f) Tangential curvature, (g) General curvature, (h) Total curvature, (i) Downslope Distance Gradient, (j) SAGA Wetness Index, (k) Terrain Ruggedness Index, (l) Stream Power Index, (m) Channel Network Base Level, (n) Vertical Distance to Channel Network.** Variables (a), (c), (d), (j), (k), (m), and (n) are displayed in the full range. Variables (b) and (l) are displayed by applying a stretching factor of 0.5 standard deviations for a better visual representation. Variables (e), (f), (g), (h), and (i) are displayed by applying a stretching factor of 2.5 standard deviations. Data source: all the maps are derived from the Digital Terrain Model 5x5, Ed. 2014 (© Archivio Cartografico, Regione Emilia-Romagna; https://metasfera.regione.emilia-romagna.it/ricerca_metadato?uuid=r_emiro:2016-08-08T155835). Maps produced using the SAGA software (version 8.1.1).

added to the list of feature variables used in the modelling procedure, as well as the lithology and land use shapefile data (Fig 7). Basics statistics including min-max range, mean, and standard deviation for each raster variable are reported in S2 Appendix (Supporting Information).

The geological units have been subdivided into 5 groups based on lithological characteristics as reported in Table 3 and Fig 7h. In particular, the *Silicified Calcilutites and Silty Clays* group is well extended in the study area, especially in the right-side of the watershed, covering a total of 19.4 km². *Varicoloured Clays and Shales* group extends only for 4.9 km² and is limited to the left-side of the watershed. *Carbonate Turbidites* group extends for 46.4 km² and it is widespread in the whole watershed. *Ophiolitic/Sedimentary Breccias and Olistoliths* group extends for 5.2 km² and is limited to the upper parts of the watershed. Finally, *Arenaceous-Pelitic Turbidites* group is extended for 12.7 km².

Land use classes have been subdivided into 10 unified LULC types as reported in Table 4 and Fig 7i. The LULC types with the related extent in the study area are: I) *Coniferous forests*, 3.8 km², II) *Beech forests*, 13.8 km², III) *Oak, hornbeam*

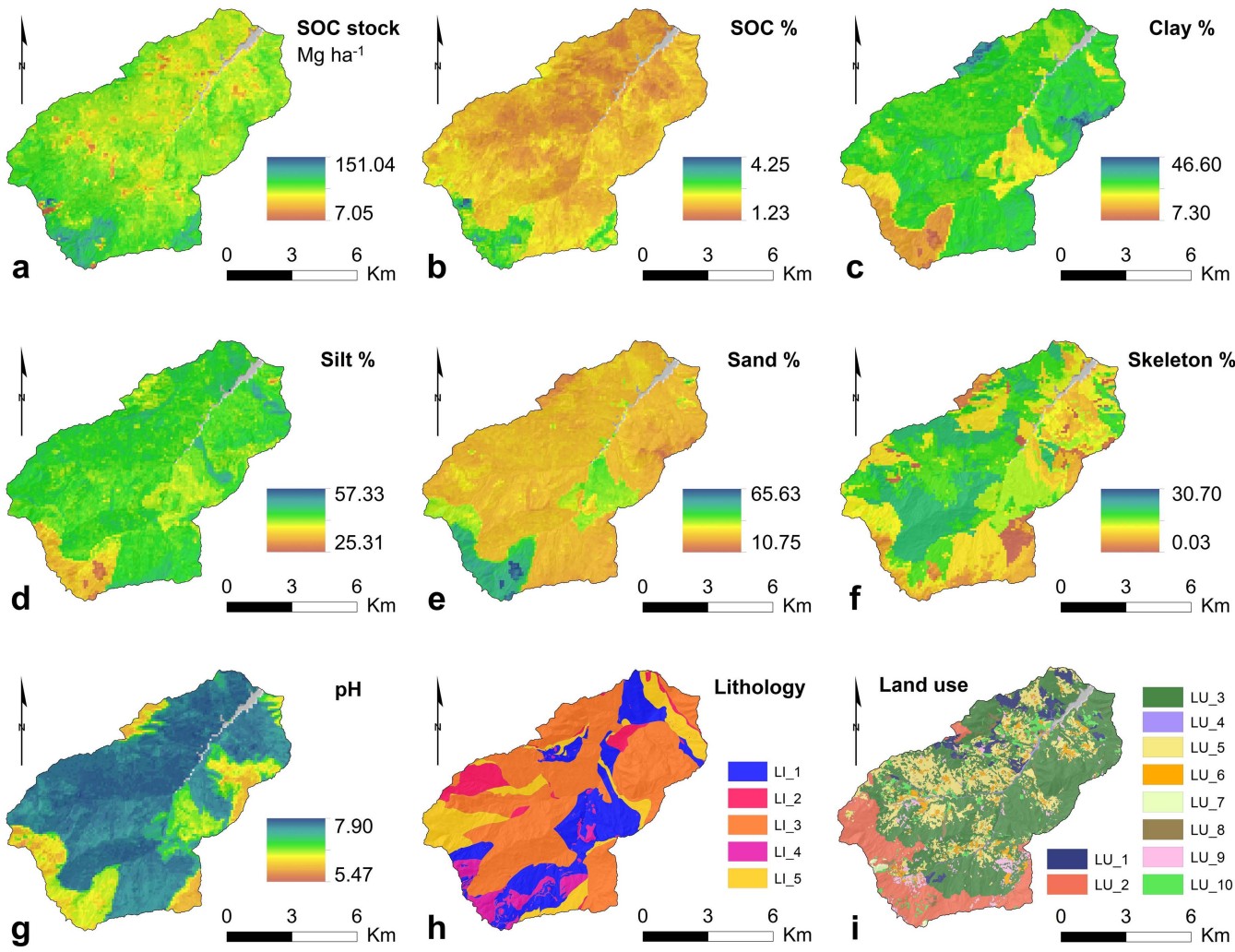

**Fig 7. Soil properties of the upper Val d'Arda-Mignano watershed: (a) SOC stock, (b) SOC %, (c) Clay %, (d) Silt %, (e) Sand %, (f) Skeleton %, (g) pH, (h) lithological groups, and (i) LULC types.** All the variables are displayed in the full range. (h) Lithological groups (Table 3): LI_1, Silicified Calcilutites and Silty Clays; LI_2, Varicoloured Clays and Shales; LI_3, Carbonate Turbidites; LI_4, Ophiolitic/Sedimentary Breccias and Olistoliths; LI_5, Arenaceous-Pelitic Turbidites. (i) LULC types (Table 4): LU_1, Coniferous forests; LU_2, Beech forests; LU_3, Oak, hornbeam and chestnut forests; LU_4, Riparian vegetation; LU_5, Cultivated fields; LU_6, Anthropic areas; LU_7, Bare surfaces/Rocky outcrops; LU_8, Sparse vegetation in evolution; LU_9, Meadows; LU_10, Bushes and shrubs. Data sources: (a) https://datacatalog.regione.emilia-romagna.it/catalogCTA/dataset/r_emiro_2023-08-09t162508; (b) https://datacatalog.regione.emilia-romagna.it/catalogCTA/dataset/r_emiro_2023-08-09t160758; (c, d, e, f) https://datacatalog.regione.emilia-romagna.it/catalogCTA/dataset/r_emiro_2023-08-02t140310; (g) https://datacatalog.regione.emilia-romagna.it/catalogCTA/dataset/r_emiro_2023-08-02t144815; (h) https://datacatalog.regione.emilia-romagna.it/catalogCTA/dataset/r_emiro_2017-06-12t113543; (i) https://geoportale.regione.emilia-romagna.it/catalogo/dati-cartografici/pianificazione-e-catasto/uso-del-suolo/layer-9 [106–111]. Maps produced using the SAGA software (version 8.1.1).

and chestnut forests, 42.2 km$^2$, IV) *Riparian vegetation*, 0.7 km$^2$, V) *Cultivated fields*, 15.7 km$^2$, VI) *Anthropic areas*, 3.7 km$^2$, VII) *Bare surfaces/Rocky outcrops*, 0.4 km$^2$, VIII) *Sparse vegetation in evolution*, 0.9 km$^2$, IX) *Meadows*, 2.6 km$^2$, X) *Bushes and shrubs*, 4.0 km$^2$.

Overall, these data represent a substantial set of environmental variables and watershed characteristics that potentially influence the geospatial variability of organic matter characterization indices used as potential indicators of erosion susceptibility.

## 4.3 Data extraction from feature variables across the experiments

The data extraction procedure across the three experiments have provided three different datasets, all of which encompass 73 rows (i.e., samples), 4 fields containing the four FTIR indices (i.e., target variables; Table 5) and 40 fields containing data extracted from the feature variables (Figs 6 and 7). The results of the data extraction from the feature variables, for the three experiments, are reported within the related datasets at the following link: https://doi.org/10.5281/zenodo.15097417 (they will not be part of the discussion).

## 4.4 Random Forest models' performance evaluation

We employed three different modelling strategies (i.e., Experiments 1–3; Fig 4) for relating different hydrological/geomorphic settings to FTIR-derived indices used as indicators of soil erosion susceptibility (Table 1).

Experiment 1 provided the best performance for the A/B index, in terms of model accuracy and agreement between predicted and observed values ($R^2$=0.39, CCC=0.59, NSE=0.38), followed by Experiment 3 ($R^2$=0.35, CCC=0.53, NSE=0.35). Experiment 2 showed slightly lower predictive power ($R^2$=0.33, CCC=0.53, NSE=0.31) (Table 7). For the A/D index, Experiment 1 similarly exhibited the best model performance in terms of predictive power ($R^2$=0.45, CCC=0.62, NSE=0.45), followed by Experiment 2 ($R^2$=0.37, CCC=0.54, NSE=0.37). Experiment 3 showed the lowest performance ($R^2$=0.32, CCC=0.48, NSE=0.32) (Table 7). For the B/D index, Experiment 3 performed better ($R^2$=0.35, CCC=0.50, NSE=0.35) than Experiment 1 ($R^2$=0.32, CCC=0.50, NSE=0.32) and Experiment 2 ($R^2$=0.33, CCC=0.47, NSE=0.33) (Table 7). For the C/D index, Experiment 1 showed the best performance ($R^2$=0.39, CCC=0.57, NSE=0.39), while Experiment 2 and Experiment 3 produced lower but comparable performance ($R^2$=0.38, CCC=0.56, NSE=0.38) (Table 7).

Considering MSE, RMSE, and Bias, for the A/B index, the models consistently exhibited similar performance across all three experiments (Table 7). Notably, the MSE remained at 0.00, the RMSE was 0.01, the Bias was also 0.00. This consistency implies that the models displayed uniform predictive accuracy and minimal error for this index across the three experimental settings. Similarly, for the A/D index, the models also achieved an MSE of 0.00, an RMSE of 0.01, and a Bias of 0.00, indicating comparable model performance (Table 7). However, for the B/D index, the models showed a higher RMSE of 0.07 across all three experiments, with a Bias of 0.00 and an MSE of 0.00 (Table 7), suggesting that while the error was slightly elevated, the model predictions remained unbiased. For the C/D index, Experiment 1 yielded a slightly higher MSE of 0.13, an RMSE of 0.36, and a Bias of 0.01 (Table 7), indicating a minor degradation in performance

Table 7. Models' performance metrics for the three experiments (cfr. Fig 4).

| FTIR Index | Experiment | R-squared | CCC | MSE | RMSE | Bias | NSE |
|---|---|---|---|---|---|---|---|
| $\frac{A}{B}$ | 1 | 0.39 | 0.59 | 0.00 | 0.01 | 0.00 | 0.38 |
| | 2 | 0.33 | 0.53 | 0.00 | 0.01 | 0.00 | 0.31 |
| | 3 | 0.35 | 0.53 | 0.00 | 0.01 | 0.00 | 0.35 |
| $\frac{A}{D}$ | 1 | 0.45 | 0.62 | 0.00 | 0.01 | 0.00 | 0.45 |
| | 2 | 0.37 | 0.54 | 0.00 | 0.01 | 0.00 | 0.37 |
| | 3 | 0.32 | 0.48 | 0.00 | 0.01 | 0.00 | 0.32 |
| $\frac{B}{D}$ | 1 | 0.32 | 0.50 | 0.00 | 0.07 | 0.00 | 0.32 |
| | 2 | 0.33 | 0.47 | 0.00 | 0.07 | 0.00 | 0.33 |
| | 3 | 0.35 | 0.50 | 0.00 | 0.07 | 0.00 | 0.35 |
| $\frac{C}{D}$ | 1 | 0.39 | 0.57 | 0.13 | 0.36 | 0.01 | 0.39 |
| | 2 | 0.38 | 0.56 | 0.13 | 0.37 | 0.00 | 0.38 |
| | 3 | 0.37 | 0.56 | 0.14 | 0.37 | 0.00 | 0.37 |

compared to other indices. In contrast, Experiments 2 and 3 for the same index produced MSE values of 0.13 and 0.14, RMSE of 0.37, and Bias of 0.00 (Table 7). This suggests that the model performed similarly in Experiments 2 and 3 but slightly better in Experiment 1 based on the MSE and RMSE metrics.

Hence, Experiment 1 performed generally better than the other two experimental designs, demonstrating that the entire upstream hydrological contributing areas (Fig 4a) provide the best way for investigating the interplay between SOM composition and selected feature variables in this landscape setting (Table 7).

## 4.5 Feature importance and variable interactions

In this section we report the results for the best performing experiment for each FTIR index (Table 7): A/B, Experiment 1; A/D, Experiment 1; B/D, Experiment 3; C/D, Experiment 1. Fig 8 shows permutation based on feature importance, while ALE plots are provided in Figs 9–12. Additionally, the feature importance permutation plots and ALE plots for all the experiments are included in Supporting Information (S1 and S2 Figs).

In general, the results provided valuable insights into the effectiveness of various environmental factors in predicting geospatial variability of SOM composition in the upper Val d'Arda-Mignano watershed.

The most important variable influencing A/B index is *Oak, hornbeam, and chestnut forests*, followed by *Vertical Distance to Channel Network* (Fig 8a), emphasizing the significant role of vegetation type and proximity to the valley bottom (i.e., main channels) in shaping SOM hydrophobic and hydrophilic characteristics. In particular, ALE plots reveal a noticeable increase in A/B as the presence of oak, hornbeam and chestnut forests approaches 0.30 of proportion in contributing areas, indicating that this forest type has a strong positive influence on SOM hydrophobicity (Fig 9), while over 0.50 the effect of this variable tends to level off. Similarly, ALE plots show a steady increase in A/B with increasing vertical distance from the channel network, especially between 200 and 300 m (Fig 9). *Carbonate Turbidites* and *Aspect* show moderate influence in the prediction of A/B index (Fig 8a), with ALE plots suggesting that the relationship between mineral and SOM composition, as well as for slope orientation, has a relatively minor and inconsistent influence compared to other features (Fig 9). *Channel Network Base Level*, *Bare surfaces/Rocky outcrops*, *Beech forests*, *Sand %*, *Clay %*, and *Elevation* show relatively lower importance (Fig 8a).

*Vertical Distance to Channel Network* is the most significant variable also in predicting A/D index, followed by *Oak, hornbeam and chestnut forests*, pointing out the critical role of topography and forest type in influencing hydrophobicity (Fig 8b). In particular, ALE plots show a steady increase in A/D as the vertical distance from the channel network increases above 250 m, whereas also in this case the positive effect of oak, hornbeam and chestnut forests emerges over 0.30 of proportion in the contributing areas (Fig 10). *Clay %*, *Channel Network Base Level*, and *Carbonate Turbidites* also appear to have a relevant role in influencing A/D prediction by the RF model (Fig 8b). Particularly, there's a significant increase in A/D at high clay percentages (> 30%), as well as at high proportion of carbonate turbidites (> 0.90), highlighting the significant effects of soil texture and lithological characteristics (Fig 10). Conversely, there is a slight decrease in A/D as the channel network base level increases above 600 m (Fig 10).

Concerning B/D index, *Elevation* and *Channel Network Base Level* emerge as the most influential variables (Fig 8c), emphasizing that elevation gradient and proximity to erosional base level strongly influence the spatial distribution of the index. Particularly, Fig 11 shows a sharp decline in index values around 500 m of elevation, followed by a moderate decrease until a threshold is reached (around 800 m). Above 800 m, the effect on the index tends to level off and becomes more stable, exhibiting a slightly negative effect (similarly to the channel network base level; Fig 11). *SOC %* is also an important variable (Fig 8c), showing a strong positive effect below 1.75%, followed by a sharp decrease as the percentage of SOC increases (Fig 11). Conversely, *Oak, hornbeam, and chestnut forests*, which also has a valuable effect on B/D index (Fig 8c), exhibits a negative effect below 0.25 of proportion in the contributing areas, followed by a moderate increase between 0.25 and 0.50 (Fig 11). Other variables such as *SOC stock*, *Beech forests*, *Downslope Distance Gradient*, and *Aspect* are progressively decreasing in importance (Fig 8c).

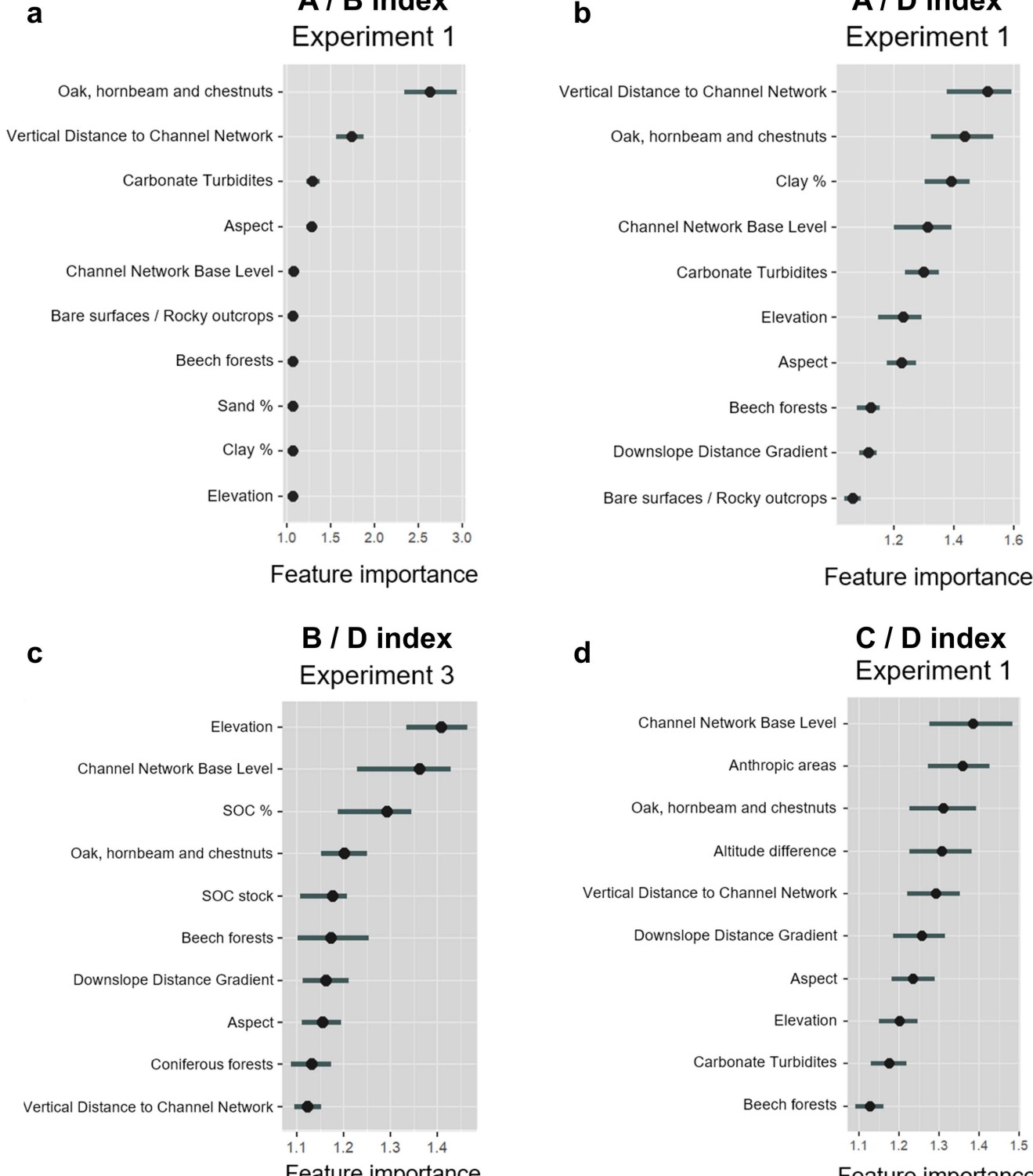

**Fig 8. Permutation-based variable importance for the selected variables in the best-performing experiment for each FTIR index: (a) A/B, Experiment 1; (b) A/D, Experiment 1; (c) B/D, Experiment 3; (d) C/D, Experiment 1.**

# A / B index    Experiment 1

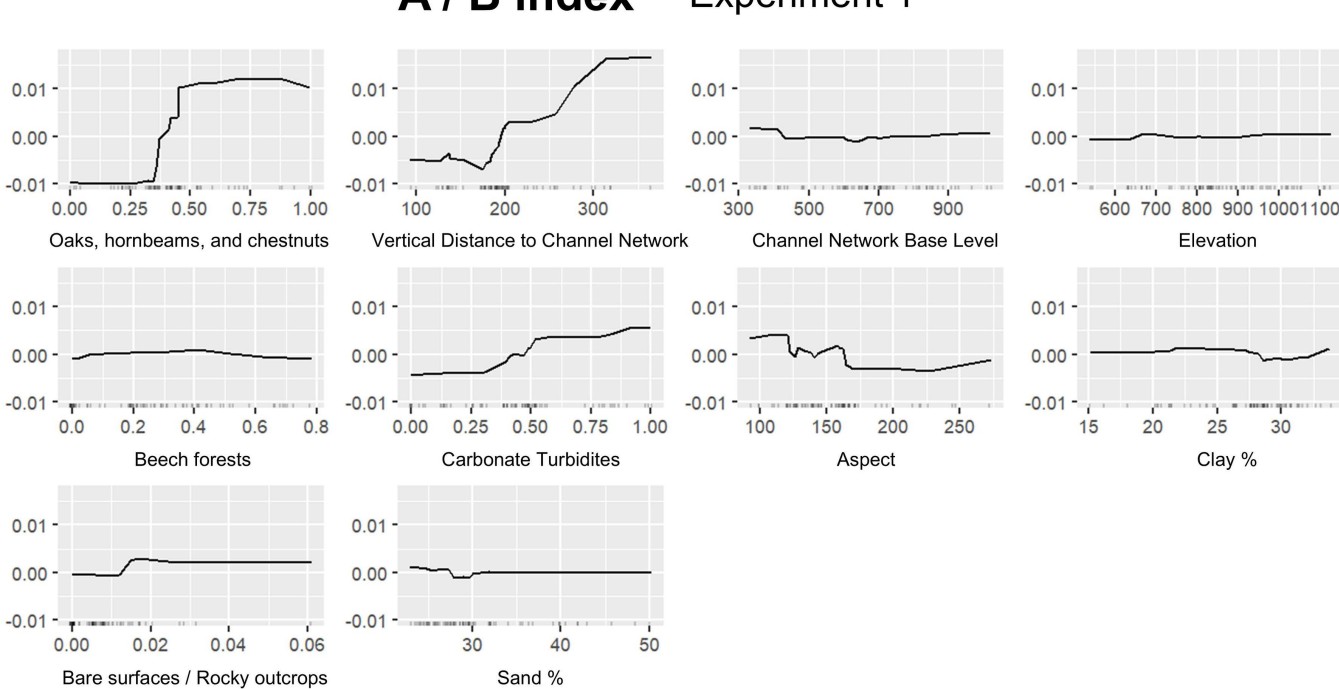

**Fig 9. Accumulated Local Effect (ALE) plots using the RF model for assessing A/B index, Experiment 1.**

# A / D index    Experiment 1

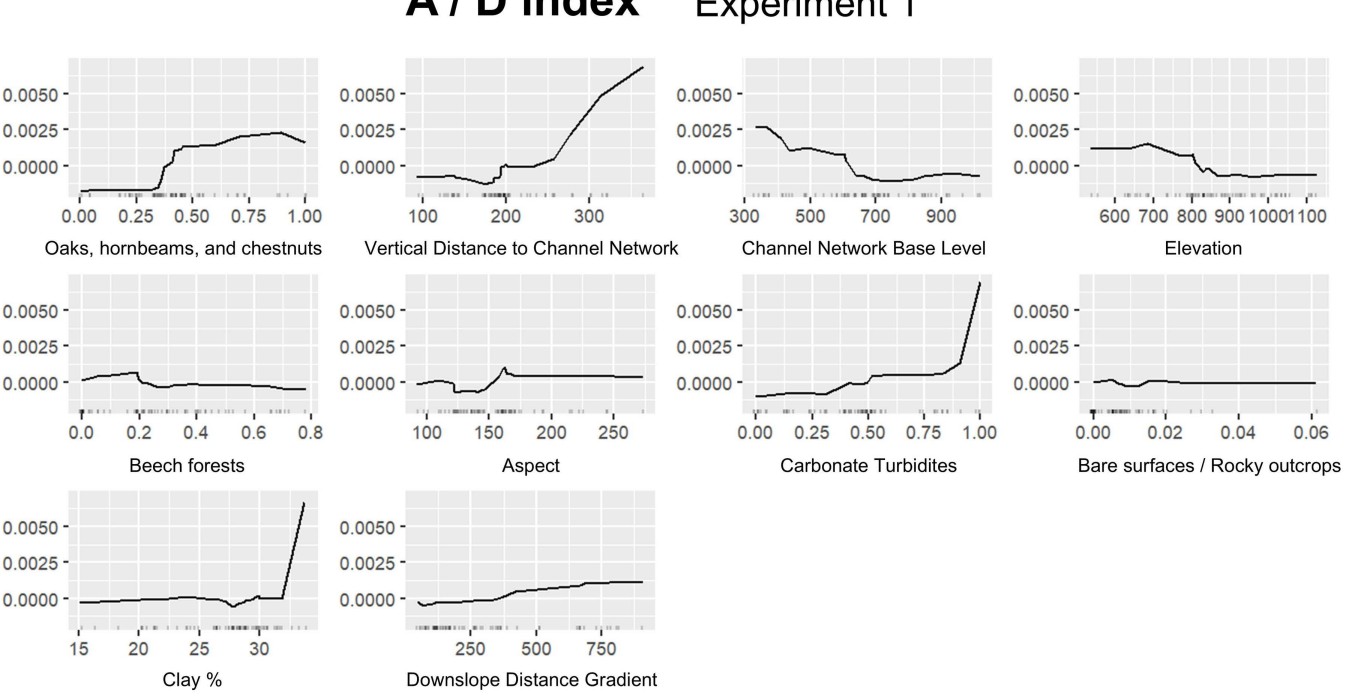

**Fig 10. Accumulated Local Effect (ALE) plots using the RF model for assessing A/D index, Experiment 1.**

# B / D index     Experiment 3

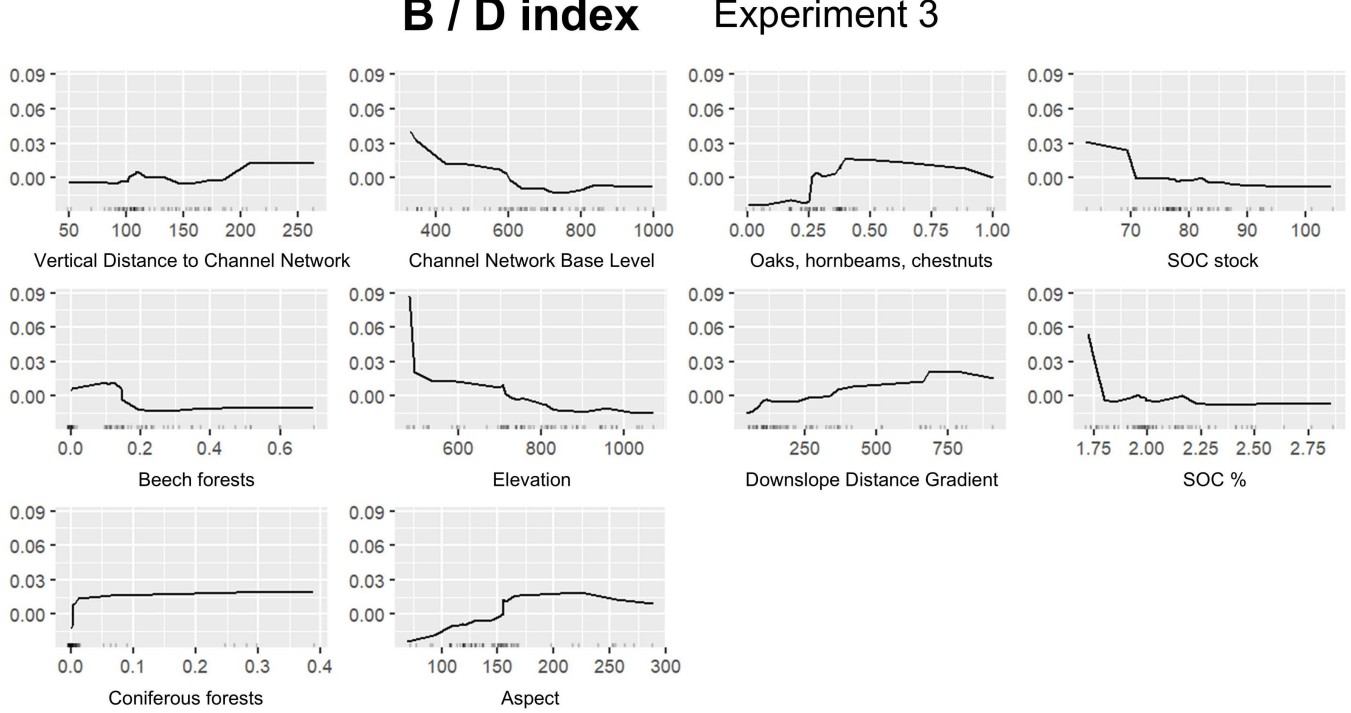

**Fig 11. Accumulated Local Effect (ALE) plots using the RF model for assessing B/D index, Experiment 3.**

*Channel Network Base Level* has the highest importance for C/D index (Fig 8d), where ALE plots highlight a strong positive effect of the proximity to erosional base level on carbonate content below 500 m, as well as a substantial negative effect above 600 m (Fig 12). *Anthropic areas* is the second most important variable (Fig 8d), exhibiting a significant increase as the proportion of the variable increases (Fig 12), suggesting that human activities have a significant effect on carbonate content, as well as on SOM composition and its interaction with cations like calcium. Also in this case *Oak, hornbeam, and chestnut forests* is a critical variable, followed by *Altitude difference*, *Vertical Distance to Channel Network*, *Downslope Distance Gradient*, and *Aspect* (Fig 8d). Notably, also in this case *Elevation* exhibits a threshold of 800 m, which separates the positive effect of elevation at lower altitudes and its negative effect at higher ones (Fig 12).

## 5 Discussion

### 5.1 FTIR-derived indices as indicators of soil degradation and erosion processes

The results of the RF model performance evaluation suggest that considering broader areas widely influenced by surface hydrological and geomorphic processes (Experiment 1), rather than focusing solely on sediment-related landforms and associated processes (Experiments 2 and 3), offers a more valuable framework for modelling indicators of soil erosion susceptibility in the upper Val d'Arda (Fig 4; Table 7). This is particularly evident for the two proxies representing the relative hydrophobicity of SOM and for the organic matter-cation associations (Table 1). It is important to note that Experiment 2 limits the contributing areas (CAs) of the sampling points to regions specifically characterized by mapped geomorphic features, including areas impacted by rill-interrill erosion, various types of landslides, badlands, gullies, and fluvial erosion [62,113], whereas Experiment 3 focuses exclusively on hazard-prone areas [62]. However, fluvial sediment collected during the sampling campaign appear to not reflect very well these processes (Fig 1b). Thus, we hypothesize that this evidence might be linked to the type of erosion process captured in the sampling. Sheet erosion was likely the

# C / D index    Experiment 1

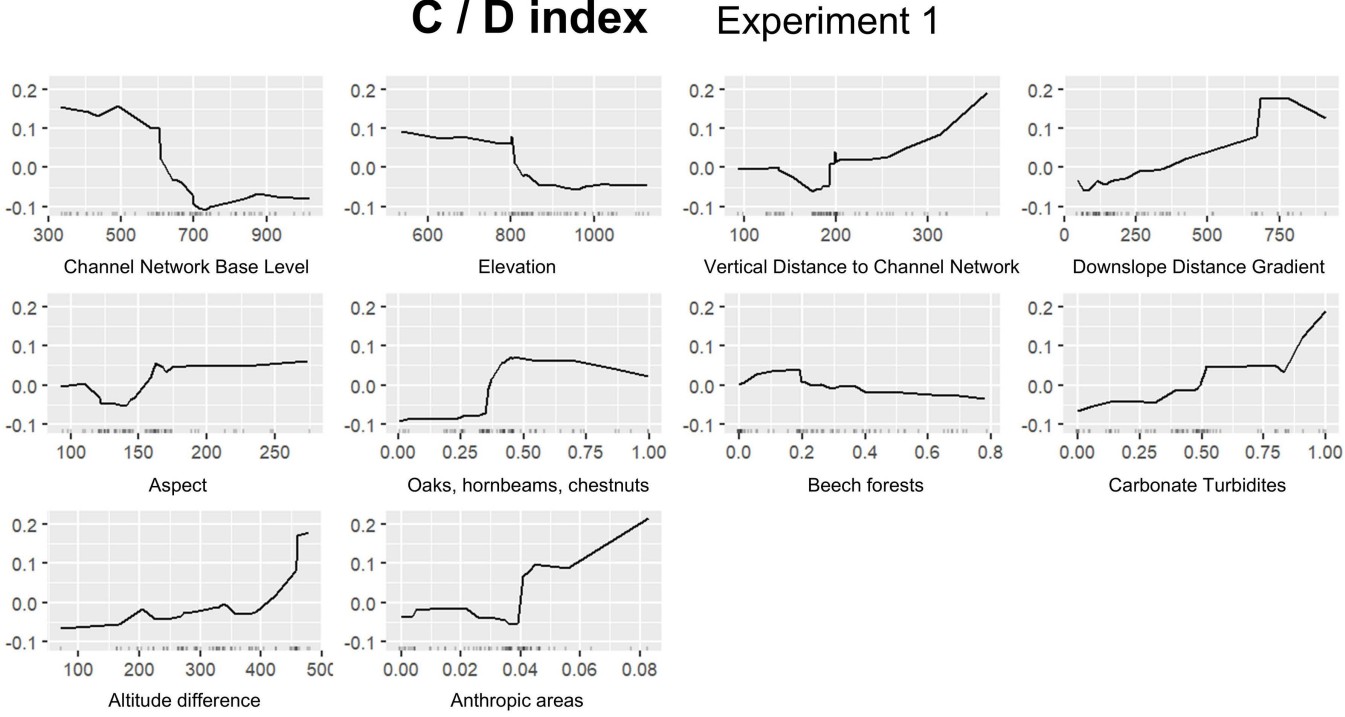

**Fig 12. Accumulated Local Effect (ALE) plots using the RF model for assessing C/D index, Experiment 1.**

dominant erosion process during the heavy rainfalls occurred in the upper Val d'Arda in May 2023, pouring out sediment and organic matter from upland areas into the drainage system. Although the watershed experienced severe sediment supply into the rivers, the hydro-meteorological event occurred in May 2023 likely impacted less on geomorphic activity than many other watersheds in the same region [i.e., Emilia-Romagna; see 126–128]. Indeed, no significant landslide (re) activations, hydro-geological hazards, or floods were observed by the Authors or reported by local authorities in the watershed [see also 75,76]. Hence, considering that diffuse sheet erosion is often difficult to detect in the field or from aerial images, especially under tree canopy, areas affected by this process could not have been adequately covered in the maps used for Experiments 2 and 3 [62] (Figs 4b, c). This might explain why this process is better represented in CAs of Experiment 1 (Fig 4a), particularly in forested areas of the watershed (Fig 7i).

Conversely, relative CEC is better predicted in hazard-prone hotspots affected by land degradation, characterized by relatively high geomorphic potential and sediment connectivity [62] (Experiment 3; Fig 4c; Table 7). Hence, our results suggest that the B/D index could be a valuable proxy for modelling soil degradation by geomorphic activity in most vulnerable areas of the watershed. Such degraded areas are probably characterized by a reduced CEC, due to the degradation of topsoil rich in SOM and organo-mineral fine fractions caused by erosion and associated processes [129,130]. Other studies previously underscored a link between reduced CEC levels, low SOM content, and increasing erosion rates in Mediterranean environments, highlighting the critical role of organic matter for soil structural stability and erosion prevention [131,132]. From a certain point of view, this evidence imply that diffuse erosion processes might have affected areas already impacted by other geomorphic processes, as more susceptible to further degradation. However, this evidence appears to contrast with the results obtained from the other three indices (Table 7), as it suggests that hazard-prone hotspots played a more important role in sediment production and export during the study period [62]. As a possible explanation, we hypothesize that the organic matter analysed in the individual samples, whose signal is

reflected in the calculation of the various indices (Tables 1 and 5), may originate from diverse areas within the respective CA. Soluble Dissolved Organic Matter (DOM) might have a different origin influenced by various factors and may have followed distinct pathways compared to organic matter in particulate form and/or associated with sediments, while the latter could have been transported differentially based on the characteristics of soil particles [133]. Moreover, Berhe [134] observed that the decomposition rates of organic matter differ considerably along dynamic toposequences and across soil depth gradients, where the effect of transfer of topsoil and associated SOM from eroding to depositional landforms, as well as of its burial by subsequent erosion events, depends on local environmental conditions. This could have implications for the role of CEC proxy as potential indicator of land degradation as, generally, the greater the degree of decomposition of the SOM, the higher the CEC of the SOM [135]. Nadeu et al. [136] already demonstrated how different land use and geomorphic settings can affect sediment sourcing and SOC dynamics in Mediterranean watersheds. Nevertheless, this study was not aimed at conducting a detailed molecular chemical analysis of organic matter to determine its origin within the watershed, nor at performing a pedological analysis to investigate soil properties in relation to degradation dynamics. Thus, the approach employed in this study is unable to offer more detailed insights into the observed relationships. However, these preliminary findings serve as a crucial starting point for further in-depth studies.

## 5.2 Geospatial relationships between target and feature variables

*Oak, hornbeam, and chestnut forests* was a recurrent critical variable for all the indices (Fig 8). This relationship might partially explain why all the investigated FTIR indices display an increasing trend in their values from upper to lower parts of the watershed (Fig 5). That is, in the Northern Apennines this forest type is typically distributed at lower-middle elevations (Fig 7i; i.e., hilly vegetation belt), whereas it is replaced by beech forests in upper areas (Fig 7i; i.e., montane vegetation belt) [137]. However, the effect of oak, hornbeam, and chestnut forests seems to be more pronounced for the relative hydrophobicity of SOM and organic matter-cation associations, as the proportion of this forest type increases, values of the related indices also significantly rise (Figs 9, 10, and 12).

On the one hand, this may imply that the organic matter inputs from this forest type contribute to increasing hydrophobic compounds in SOM relative to hydrophilic ones, by providing hydrocarbons, triglycerides, lipids, or waxes [22,138]. Forests generally provide important ecosystem services protecting soil from surficial water erosion [e.g., stabilization of soil, increasing organic matter content, and protection from rain drops erosive force; [139]. However, under the same conditions, hydrophobicity may stabilize or destabilize soil aggregates depending on soil texture and SOM content [19,23], thus, leading to a differential susceptibility to erosion. In particular, our results indicate that high clay content (i.e., *Clay%* > 30–32%) significantly improves the accumulation of hydrophobic SOM relative to polysaccharides (i.e., A/D index; Fig 8b), whereas the influence of this variable is negligible at lower concentrations (Fig 10). This may imply that once a critical amount of clay is present in the soil, it begins to have a more pronounced effect on the retention of hydrophobic compounds into organo-mineral complexes, although they produce low energy bonds that may be reversible [140,141]. Anyway, in many Mediterranean watersheds such as the upper Val d'Arda, clay-rich soils can experience intensified shrinking-swelling phenomena which enhance soil cracks under dry conditions [47,142,143]. Bosino et al. [144] highlighted that after long dry periods, these cracks may allow huge water infiltration into the upper topsoil horizon, leading to a subsequent closure of the fissures in the surface crust. During moist conditions the clay rich substrate enhance rapid runoff processes due to reduced infiltration rates, especially in shallow soil profiles with minimum storage capacities [144]. However, structural weaknesses in high-hydrophobic soils can create preferential flow paths [145], leading to uneven wetting, aggregate breakdown, and increased erosion rates under heavy rainstorms. Nevertheless, other studies pointed out that high organic matter content can stabilize clays reducing swelling and crack formation [146]. Therefore, our findings might have relevant implications for Mediterranean watersheds that are similar to the upper Val d'Arda, where clay content in soils ranges from 7 to 47% (Fig 7c), with a mean value of ~ 30% (cfr. S2 Appendix).

On the other hand, in the case of C/D index (Figs 8d and 12), the degree of decomposition or humification of the litter produced by oak, hornbeam, and chestnut trees, as well as the relative carbonate content in the samples, might provide specific compounds responsible for improving associations between organic matter and cations [39,135] (Table 1), thus promoting soil aggregate stability [19].

SOM hydrophobic characteristics appear to be largely affected also by *Vertical Distance to Channel Network* (VDCN) (Fig 8a, b). Following Berhe & Kleber [133] and Maerker et al. [102], the redistribution of SOM throughout the watershed driven by water fluxes, as affected by the topographic setting and landform position in a toposequence, could lead to a differential stabilization of the organic matter, depending on whether it is transferred through eroding areas or accumulated in depositional environments [147,148]. In this case, the relative amount of hydrophobic vs. hydrophilic components measured in our fluvial sediment samples might reflect the organic matter connectivity between the respective contributing areas and the fluvial system [149]. That is, hydrophilic compounds, being more soluble in water, are more easily transferred as DOM in areas closer to main channels (in vertical distance), where runoff volume is typically higher and flooding is more frequent and intense. Therefore, the retention of DOM within riparian vegetation or floodplain wetlands [149] might favour the accumulation of hydrophilic components such as C=O and C-O-C groups, as suggested by the negative effect of VDCN on the A/B and A/D indices below 200 m (Figs 9 and 10). This allows hydrophobic compounds to be retained in poorly connected areas, farther away from the valley bottom (in vertical distance) (Figs 9 and 10), thereby rendering soils more susceptible to water erosion by affecting water infiltration capacity and surface runoff.

Similarly, the spatial patterns of CEC and organic matter-cation associations are highly affected by topography, as reflected by the *Channel Network Base Level* (CNBL) (Fig 8c, d). Indeed, Figs 11 and 12 show that CNBL has a particular effect on these indices, where their values decrease as the erosional base level increases (see also Figs 5 and 6 m). Therefore, our results suggest that areas with a lower CNBL (closer to the reservoir) tend to accumulate more carboxylic acids, as well as more carbonates relative to silicates [84,87]. Carboxylic acids are likely more stable and exhibit stronger interactions with cations, as C=O functional groups are characteristic of more decomposed and reactive organic matter [e.g., humified materials and lignin degradation products; 50]. This suggests that as organic matter moves through the watershed, it alternates between phases of storage and remobilization depending on hydrological conditions, undergoing successive stages of decomposition that enhance its chemical reactivity [87], particularly when buried [134]. Therefore, long retention times under certain hydrological conditions might promote the formation and accumulation of carboxyl groups derived from decomposition processes, thus promoting the formation of stable organo-mineral complexes. This implication is supported also by the influence of the *Downslope Distance Gradient* (DDC) (Fig 8c, d), as areas with higher DDC values, such as in the lower watershed (see Fig 6i), appear to have a positive effect on the indices (Figs 11 and 12). Anyway, as with many other watersheds in the Northern Apennines, in the study area the past and current riverbed evolution is a function of fluvial morphodynamics, as well as lithological and structural characteristics of the landscape [61]. La Licata et al. [47] emphasised that in the lower part of the upper Val d'Arda, where gentle open slopes composed of pelitic and chaotic rock formations prevail and the valley bottom is wider, slope and fluvial sediments can be easily stored in depositional storages. According to Doetterl et al. [150], these findings underscore the importance of adopting a landscape-based modelling approach for assessing the complexity of SOM dynamics and the relations with their shaping factors, informing useful predictions of organic matter cycling and soil carbon sequestration in degraded watersheds.

In this regard, particular attention should be paid to B/D index (Fig 8c). Even though the influence of *Elevation* is more or less relevant in all experiments (Fig 4; cfr. Feature Importance plots in S1 Fig), the peculiar effect revealed by Experiment 3 is likely related to the distribution of geomorphologically highly active areas (e.g., landslide deposits connected to the drainage system and affected by additional processes like surficial soil erosion and/or stream incision, and steep rock walls affected by rockfalls and localized debris flows) with respect to the elevation gradient (Fig 11; Fig 4c) [see 62]. In other words, highly active sediment sources and eroding steep slopes, mostly distributed at medium-high elevations, likely retain less SOM rich in C=O groups because the export of organic matter is higher and the accumulation of decomposed

compounds is limited [134]. Nonetheless, the Experiment 3 could have biased the effect of SOC% on the B/D index, as in the other experiments SOC content seems to not to play an important role in shaping CEC. Indeed, generally high SOC content improves CEC [16], with a consequent effect on soil aggregate stability [19]. However, Fig 11 shows a significative positive effect of SOC content only for low percentage (around 1.75%), while the effect of increasing SOC content appears to be negligible. This evidence may indicate that in geomorphologically active degraded areas, a minimum threshold of SOC is needed to contribute to CEC. However, the continued disturbance and removal of organic matter in these areas may be overriding the beneficial effects of increasing SOC content (Fig 11).

Human-altered landscapes (i.e., *Anthropic areas*; Fig 7i; Table 4), including urban network, mining areas, construction sites, excavations, manufacturing facilities, infrastructures, farming and livestock settlements, and agricultural areas (other than cultivated fields), seem to promote a greater degree of organic matter-cation interactions (i.e., C/D index; Fig 8d). This could be due to soil amendments (e.g., fertilizers) or certain land management practices that enhance nutrient retention and cation exchange processes. Moreover, carbonate-rich soils (i.e., *Carbonate turbidites*; Fig 7h) appear to have a positive effect on C/D index (Fig 12), likely due to the presence of calcium and other cations, which improve soil stability. This finding is explained by the widespread distribution of carbonate-rich lithologies in the contributing areas of the analysed samples in Experiment 1 (Fig 4a). Particularly, it is worth noting that samples with absorption spectra showing a high-intensity band C (i.e., higher or similar compared to band D; Fig 3) have a mean proportion of *Carbonate Turbidites* of 0.55 (C/D > 1) and 0.41 (C/D close to 1), which is quite higher than other lithologies (Table 3; see available Datasets). Thus, these results underscore the relation between C/D index and the relative carbonate content in the samples (Table 1) [53,55].

## 5.3 Limitations of the methodological approach and improvements

Despite these promising findings, certain limitations need to be acknowledged.

In general, the significance of these findings relies on the assumption that the FTIR indices used in this study can serve as proxies for soil hydrophobicity, CEC, and organic matter–cation interactions (Table 1) within this environmental context, as supported by evidence drawn from the literature [39,50–55,84]. The obtained results represent an initial preliminary attempt towards a better understanding of such relationships. However, these results should be interpreted with caution, e.g., as we discussed for their relevance in terms of erosion processes. That is, an empirical evaluation linking FTIR indices to actual erosion measurements—necessary to establish these indices as direct indicators—is currently lacking. Nonetheless, additional studies are needed to explore more detailed relationships between these potential FTIR-based indicators and the chemical and pedological characteristics of sediment source areas.

The relatively higher errors associated with C/D, which are likely caused by interference problems (e.g., signals from SOM and soil mineral components, or effects of land use vs. soil genesis), suggest that the model's capability to predict SOM chemical characteristics with high variability (cfr. STD = 0.47) should be interpreted cautiously (Table 7). This underscores the need for further refinements in modelling approaches to better capture the complexity SOM-environment interactions, especially in areas with diverse vegetation and soil types. Considering the potential overlapping in absorbance bands between organic and mineral fractions, especially in the case of band C (Carbonate/$v_s$COO⁻) and band D (C-O-C/Si-O-Si) (Fig 3), further improvements should disentangle the effect of organic vs. mineral compounds in the spatial distribution of the indices. On the other hand, if the organic component is not separated from the mineral fraction, accounting for the particle size effect and effective organic matter content on organic matter–sediment interactions and the detectability of SOM characteristics could reveal additional insights into the variability of the measured band intensities [49,151]. This might be of particular interest whether the sediment's particle size distribution and/or organic constituents allow for hypotheses regarding the origin of the sediment within the watershed or enable inferences about the conditions and the selectivity of the mechanism responsible for its erosion and transport [78,79,152,153]. Anyway, recent advancements in FTIR analysis [e.g., 154,155] suggest that characterizing silicate and carbonate components could provide

additional insights into the role of these compounds on aggregate stability and hydro-mechanical properties of soils [156]. Additionally, the reliance on FTIR indices alone may limit the range of SOM characterization, as these indices provide only a partial view on the broader ranging organic matter dynamics in the watershed (cfr. Table 7).

Furthermore, the RF model used has potential limitations, including the lack of tuning for other hyperparameters such as the number of trees (ntree) or the minimum node size, which could significantly impact model performance. Since the *rf* function in the 'caret' package only allows tuning of the 'mtry' parameter [116,117], the optimization process may not fully exploit the potential of the RF algorithm. Also, RF may struggle with extrapolation or interpretability in certain cases, and comparing its performance with other machine learning models like gradient boosting machines or support vector machines could provide insights into whether a different model better captures the relationships in the data. Despite the reliability afforded by the LOOCV and ensemble structure of RF model, residual uncertainties may persist due to potential imbalances in the datasets, the underrepresentation of specific geomorphic features where geomorphological mapping is limited by a dense forest cover, or variability in SOM composition that is not captured in the sampling. These factors may limit the model's transferability to unsampled areas having diverse environmental conditions.

Future research should focus on expanding the spatial and temporal scales of investigation to include different watersheds with varying climatic and geomorphological conditions. This would allow for a more comprehensive understanding of how SOM composition interacts with environmental factors across different contexts. Moreover, integrating additional variables such climatic data and land management practices, as well as incorporating dynamic modelling approaches, could provide further insights into the relationships between soil erosion susceptibility and SOM dynamics in Mediterranean environments.

## 6 Conclusion

With this interdisciplinary work, we provide significant new evidence on the relationships between SOM-related properties, serving as potential indicators of soil degradation and erosion, and environmental, geomorphic, and hydrological characteristics in a temperate agricultural-forested area of the Northern Apennines chosen as representative of Mediterranean environments. Particularly, our approach explored SOM-related indices as proxies for relative hydrophobicity, CEC, and the degree of organic matter-cation interactions, identifying the most influential environmental predictors driving their spatial variability at watershed scale and evaluating their effect on the observed values. Moreover, the significance of our approach exceeded the initial objectives, providing valuable new insights into the complex landscape-based relationships between environmental factors, geomorphology, connectivity, and SOM dynamics. The key findings of this study can be summarized as follows:

- Relative hydrophobicity of SOM and organic matter-cation interactions are better predicted in areas influenced by broader hydrological and low-magnitude geomorphic processes during the study period, as reflected by the entire contributing areas of the sampling points (Experiment 1), likely reflecting diffuse erosion processes like sheet erosion. This highlights the importance of considering low-intensity geomorphic regimes, beyond more intensive processes like gullying and landsliding, for assessing and managing sediment supply to river systems during hydrometeorological events, particularly in forested areas;

- CEC was identified as a proxy for detecting and modelling soil degradation caused by geomorphic activity in the most vulnerable areas of the watershed, as reflected by the hazard-prone contributing areas used in Experiment 3. This evidence partially contrasts with the other indices, suggesting that diffuse erosion may preferentially affect already degraded areas. This underscores the complexity of SOM dynamics in environments with similar characteristics and supports new research directions exploring the differential origins and transport pathways of SOM based on its chemical properties and associations with mineral particles on hillslope and sub-watershed scales;

- Areas dominated by oak, hornbeam, and chestnut forests, on carbonate-rich bedrocks, and located farther from the valley bottom, tend to have more hydrophobic soils, making them potentially more prone to water erosion. Additionally, high clay content in soil appears to favour the retention of hydrophobic compounds, potentially intensifying shrinking-swelling

phenomena, soil cracking, and localized water erosion. These findings carry significant implications for Mediterranean watersheds having similar vegetation, as well as for regions with seasonally contrasting climates and lithologies dominated by weak, stratified, clay-rich rocks;

- Topographic features such as elevation and channel network base level were found to significantly influence the spatial patterns of CEC and organic matter-cation interactions. This suggests that topography-driven erosion potential and landscape connectivity play critical roles in the redistribution and retention of organic matter across landscape compartments. These insights suggest that such landscape-based modelling could be a valuable approach for predicting organic matter cycling and carbon sequestration in complex, dynamic Mediterranean watersheds with similar landscape configuration;

- Organic matter-cation associations appear to be strongly influenced by relative carbonate content, likely due to higher concentrations of calcium and other cations, as well as specific land management practices. This has important implications for soil erosion control strategies, as stronger SOM-cation interactions enhance soil aggregate stability, reducing susceptibility to erosion.

While providing significant findings that support new research hypotheses and perspectives, at this stage, our interdisciplinary approach is able to explain only a moderate portion of the variability in the SOM indices, indicating the need for further improvements to enhance model performance. Future studies could focus on implementing broader spatial and temporal analyses, integrating climatic and land management variables. Additionally, more refined geostatistical methods could be applied to spatially model and regionalize soil erosion susceptibilities by integrating the most influential predictive features.

## Supporting information

**S1 Appendix. Additional information regarding laboratory procedures and methodology.**
(PDF)

**S2 Appendix. Basic statistics for the raster variables selected as feature variables.**
(PDF)

**S1 Fig. Permutation-based variable importance for all the experiments.**
(PDF)

**S2 Fig. Accumulated Local Effect (ALE) plots for all the experiments.**
(PDF)

## Acknowledgments

We thank the staff of Consorzio di Bonifica di Piacenza for the valuable support during the field surveys. We were also supported by the Belmont ABRESO project.

## Author contributions

**Conceptualization:** Manuel La Licata, Odunayo D. Adeniyi, Ruth H. Ellerbrock, Michael Maerker.

**Data curation:** Manuel La Licata.

**Formal analysis:** Manuel La Licata, Odunayo D. Adeniyi.

**Funding acquisition:** Michael Maerker.

**Investigation:** Manuel La Licata, Odunayo D. Adeniyi.

**Methodology:** Manuel La Licata, Odunayo D. Adeniyi, Ruth H. Ellerbrock, Michael Maerker.

**Project administration:** Michael Maerker.

**Resources:** Ruth H. Ellerbrock, Natalie Papke.

**Software:** Manuel La Licata, Odunayo D. Adeniyi.

**Supervision:** Ruth H. Ellerbrock, Michael Maerker.

**Validation:** Manuel La Licata, Odunayo D. Adeniyi.

**Visualization:** Manuel La Licata, Odunayo D. Adeniyi.

**Writing – original draft:** Manuel La Licata.

**Writing – review & editing:** Manuel La Licata, Odunayo D. Adeniyi, Ruth H. Ellerbrock, Nisha Bhattarai, Alberto Bosino, Natalie Papke, Jörg Schaller, Michael Maerker.

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
