## [Decision Letter · Decision Letter 0]

15 May 2025

PONE-D-25-04924

Integration of FTIR-based soil degradation indices with stochastic modelling to assess spatial patterns of organic matter-sediment dynamics in a Mediterranean watershed

PLOS ONE

Dear Dr. La Licata,

Thank you for submitting your manuscript to PLOS ONE. After careful consideration, we feel that it has merit but does not fully meet PLOS ONE’s publication criteria as it currently stands. Therefore, we invite you to submit a revised version of the manuscript that addresses the points raised during the review process.

Manuscript presents a valuable and methodologically sound case study integrating FTIR-based soil characterisation with spatial modelling. Reviewer 1 recommends acceptance, while Reviewer 2 requests minor revisions regarding title precision, figure quality, and clarification of erosion processes. In addition to these points, I require the authors to address several editorial concerns:

(1) avoid overgeneralisation of results beyond the studied catchment;

(2) moderate the interpretation of FTIR indices as direct erosion proxies, clearly stating their indirect nature;

(3) discuss the potential sampling bias due to post-event sediment collection;

(4) critically reflect on model uncertainties and limitations; and

(5) better integrate lithological and geomorphological controls into the analysis.

These revisions are necessary to meet PLOS ONE’s standards of scientific transparency and accurate interpretation.

We look forward to receiving your revised manuscript.

Kind regards,

Przemysław Mroczek, Dr. hab.

Academic Editor

PLOS ONE

Journal Requirements:

“This research was conducted with the financial support of the Earth and Environmental Sciences PhD-PON program (Research & Innovation, 2014e2020, Education and research for recovery - REACT-EU, DOT1322534-4) of University of Pavia, Department of Earth and Environmental Sciences. This research was also funded by the PRIN 2022 project by the Italian Ministry of University and Research, entitled: ”Full cOveRage, Multi-scAle and multi-sensor geomorphological map: a practical tool for TerrItOrial plaNning – FORMATION” (2022C2XPK7_004). This research was also supported by the Polish National Agency for Academic Exchange - Urgency Grants program project contract no. BPN/GIN/2024/1/00008.”

6. We note that Figures 1 and 4-7 in your submission contain map/satellite images which may be copyrighted. All PLOS content is published under the Creative Commons Attribution License (CC BY 4.0), which means that the manuscript, images, and Supporting Information files will be freely available online, and any third party is permitted to access, download, copy, distribute, and use these materials in any way, even commercially, with proper attribution. For these reasons, we cannot publish previously copyrighted maps or satellite images created using proprietary data, such as Google software (Google Maps, Street View, and Earth). For more information, see our copyright guidelines: http://journals.plos.org/plosone/s/licenses-and-copyright.

 1. You may seek permission from the original copyright holder of Figures 1 and 4-7  to publish the content specifically under the CC BY 4.0 license. 

Additional Editor Comments:

Dear Authors,

Thank you for submitting your manuscript entitled “Integration of FTIR-based soil degradation indices with stochastic modelling to assess spatial patterns of organic matter-sediment dynamics in a Mediterranean watershed” to PLOS ONE. I appreciate the considerable effort invested in this interdisciplinary study, which integrates FTIR spectroscopy with geomorphological and statistical modelling approaches to investigate organic matter and sediment dynamics in a Mediterranean catchment.

The manuscript has been reviewed by two external referees. Reviewer 1 provided a very positive evaluation, stating that your research is technically sound, the data are sufficient and appropriately analysed, and the manuscript is clearly written. They commended the scientific relevance of your study and acknowledged the improvements made since the initial submission. Reviewer 2 also recognised the value of your work but raised several points requiring attention. Specifically, they suggested that the manuscript’s title should be more precise, explicitly reflecting the study area rather than implying a broader regional scope. They noted deficiencies in figure quality, particularly Figure 1, and recommended improvements in visual presentation, including the addition of geological and land use maps, as well as photographs of typical lithologies. Furthermore, they pointed out inconsistencies in figure referencing within the text and encouraged a more nuanced discussion of erosion processes, especially regarding the role of hydrophobicity in soil stability.

In addition to the reviewers' comments, I would like to draw your attention to several further issues that require careful consideration. While your integration of FTIR-derived indices with Random Forest modelling is methodologically sound, the interpretation of these proxies as direct indicators of erosion susceptibility appears overstated. The manuscript lacks empirical validation linking the FTIR indices to actual erosion measurements, sediment yields, or field-observed processes. This disconnect should be addressed with a more cautious interpretation of your findings, clearly acknowledging the proxy-based and indirect nature of your assessments.

Moreover, your conclusions frequently extrapolate results from a relatively small catchment in the Northern Apennines to “Mediterranean watersheds” in general. Such generalisations are not sufficiently substantiated and should be carefully delimited to the specific physiographic and climatic context of your study area. A more balanced formulation that frames your findings as case-study insights rather than general regional conclusions is recommended.

Another aspect requiring attention concerns the sampling strategy. Given that samples were collected following a significant rainfall event, it would be pertinent to discuss how this may have influenced sediment composition and whether such conditions are representative of the typical sediment dynamics in the catchment. This context is essential for interpreting your results.

Additionally, while your Random Forest models are well described, there is little discussion of model uncertainties, potential overfitting, or limitations. Providing a critical reflection on the robustness of your models and discussing the possible influence of dataset characteristics would enhance the scientific transparency of your work.

Finally, the integration of lithological controls and the role of mass movements in sediment connectivity and SOM dynamics are not fully explored. Although mentioned in the text, these factors are not quantitatively assessed nor sufficiently linked to your FTIR-based analyses. A more detailed examination of how different lithologies and geomorphological processes affect your observed patterns would significantly strengthen the study.

I therefore invite you to revise your manuscript by addressing both the reviewers’ comments and the editorial concerns outlined above. I believe these improvements will enhance the clarity, scientific rigour, and overall impact of your work.

Yours sincerely,

Przemysław Mroczek

Academic Editor

PLOS ONE

Reviewers' comments:

Reviewer's Responses to Questions

**Comments to the Author**

1. Is the manuscript technically sound, and do the data support the conclusions?

Reviewer #1: Yes

Reviewer #2: Yes

2. Has the statistical analysis been performed appropriately and rigorously? 

Reviewer #1: Yes

Reviewer #2: Yes

3. Have the authors made all data underlying the findings in their manuscript fully available?

Reviewer #1: Yes

Reviewer #2: Yes

4. Is the manuscript presented in an intelligible fashion and written in standard English?

Reviewer #1: Yes

Reviewer #2: Yes

5. Review Comments to the Author

Reviewer #1: I believe that authors put sufficient data in order to explain their approach and to enable reproducibility. This is a a valuable contribution for the study of landscape evolution in Mediterranean watersheds. I also understand that the MS went through one round of review with different reviewers and authors made significant changes.

Reviewer #2: The title of the article suggests research in the Mediterranean region, whose coastline is very long, extensive, and is based on many countries, regions, etc. Meanwhile, when reading the text, we read that the authors are basically studying the catchment area of one river flowing into the Adriatic Sea, which concerns the region of Emilia-Romagna, northern Italy, the Arda River. This means that the title should be changed, precisely defining the area.

Fig. 1. The quality of these images is low, it needs to be improved. The hatching also raises concerns because the maximum height of 1300m is marked as if it were a height of over 5000m.

Line 134 and others: I suggest writing Roman numerals with a capital letter.

The authors described the catchment area of the studied river, also showing the lithology of the bedrock. They indicate in the text that mass movements occur there. Maybe it would be worth including a geological and soil map in this place? It would be good to explain why mass movements occur, whether it is the effect of soil instability resulting from heavy rains, the arrangement of rocks in the subsoil, or improper use of soils. This information has an impact on the rest of the text, especially since the authors themselves indicate that mass movements destroy soils (line 149). It is also worth considering how fields are ploughed, which can facilitate or hinder surface runoff of water on slopes.

Table 3 is interesting, but the authors could include photographs of typical examples of rocks and a map of agricultural use of the research area.

After Fig. 5, the authors describe Figure 7 (line 439) in the text, so it should be moved so that the order of citation of graphics in the text is maintained. There is no description of Figure 6 in this paragraph. This needs to be supplemented. There is only one reference in line 473, but it is too little for the information contained there.

Line 571, requires consideration, the hydrophobicity of matter alone does not have to stop erosion, which can be caused by, for example, soil instability and mechanical washing out. In addition, chemical compounds can appear in polluted waters that cause emulsification of such compounds.

6. PLOS authors have the option to publish the peer review history of their article (what does this mean? ). If published, this will include your full peer review and any attached files.

**Do you want your identity to be public for this peer review?** For information about this choice, including consent withdrawal, please see our Privacy Policy .

Reviewer #1: **Yes: ** Bülent Arıkan

Reviewer #2: **Yes: ** Miłosz Huber

---

## [Author Response · Author response to Decision Letter 1]

6 Jun 2025

Dear Editor,

Dear Reviewers,

We would like to express our gratitude for your valuable comments and suggestions, which have significantly contributed to improve the manuscript. We have tracked the changes throughout the manuscript and provided a clean copy without tracked changes, as requested by the journal. We have addressed your requests and comments throughout the manuscript. Please find below our responses. The specific references to the modified lines are referred to the “clean copy”.

Reviewers’ comments:

Reviewer_1: The title of the article suggests research in the Mediterranean region, whose coastline is very long, extensive, and is based on many countries, regions, etc. Meanwhile, when reading the text, we read that the authors are basically studying the catchment area of one river flowing into the Adriatic Sea, which concerns the region of Emilia-Romagna, northern Italy, the Arda River. This means that the title should be changed, precisely defining the area.

Corresponding author (CA): Thank you so much for the valuable comment. We agree with you and modified the title accordingly. Now the title sounds: “Integration of FTIR-based soil degradation indices with stochastic modelling to assess spatial patterns of organic matter-sediment dynamics in a Mediterranean watershed – A case study from the Northern Apennines”. We believe that this title better reflect the regional specificity of the study watershed, while keeping the attention of the reader to the Mediterranean. Hence, we believe that the reader should be clearly informed from the title that we are dealing with Mediterranean characteristics, in spite of we are studying only one watershed in the Apennines. The title is 183 characters without spaces or 211 with spaces, thereby being consistent with journal requirements (i.e., max 250 characters).

Reviewer_2: Fig. 1. The quality of these images is low, it needs to be improved. The hatching also raises concerns because the maximum height of 1300m is marked as if it were a height of over 5000m.

CA: We greatly appreciate your comment and your attention to the quality of the figures. It is possible that the version of the figure displayed in the submission document does not show the overall map and the photographs below with optimal clarity. However, we have provided all images at a resolution of 500 dpi, and we are confident that the final publication will ensure high-quality rendering. If by “image quality” you referred to something else, we kindly ask you to clarify this point…

Figure 1 provides all the necessary elements to properly outline the study area in its geographical context, offering both a regional overview and a basic characterization of the topography of the area. This is intended to help the reader better understand the distribution of sampling sites across the landscape. A zoomed-in panel further illustrates the relative position of the samples along the valley floor, accompanied by representative photographs (in high resolution). Therefore, we believe that this figure offers a useful and essential tool for contextualizing the sampling design within the studied watershed. The legend clearly shows the elevation range of the study area, leaving no particular ambiguity about the topographic configuration or actual elevation of the watershed. The DTM was classified using graduated colours, based on a scale bar that is commonly used for elevation data, and semi-transparently overlaid on a hillshade map to improve the visualization of landscape morphology (an approach widely used for this type of map). Thus, we understand that the original colour scale, from light-blue (valley floors) to white (main peaks), may have been somewhat misleading at first glance, as it may have suggested higher alpine-like peaks. For this reason, we have adjusted the scale bar by removing the white colour, making the visualization more appropriate for the actual elevation range of the study area. Please check the new Figure 1 provided along with the revised submission. Anyway, please consider that Fig. 1a has been changed to accomplish journal requirements (see comment below; cfr. comment ‘Journal_6’).

Reviewer_3: Line 134 and others: I suggest writing Roman numerals with a capital letter.

CA: Thank you for the comment. Roman numerals have been rewritten using capital letters throughout the text.

Reviewer_4: The authors described the catchment area of the studied river, also showing the lithology of the bedrock. They indicate in the text that mass movements occur there. Maybe it would be worth including a geological and soil map in this place? It would be good to explain why mass movements occur, whether it is the effect of soil instability resulting from heavy rains, the arrangement of rocks in the subsoil, or improper use of soils. This information has an impact on the rest of the text, especially since the authors themselves indicate that mass movements destroy soils (line 149). It is also worth considering how fields are ploughed, which can facilitate or hinder surface runoff of water on slopes.

CA: Thank you for your suggestion and consideration. In our opinion, including geological and soil maps in the Study Area chapter would go beyond the scope of this work, considering that this study does not aim to explore relationships between erosion and landslide processes and specific variables such as all the outcropping geological formations or soil types. In this study, geological formations are grouped based on lithological characteristics and relevant/available soil properties are taken separately as single variables. Moreover, for the whole study area, the available soil map is at a scale of 1:250,000, while more detailed information (1:50,000) is only available for limited parts of the watershed. Therefore, the integration of pedological data in this form would likely not be particularly useful. The core of the study lies in the relationship between the so-called ‘target variables’ (FTIR indices) and the ‘feature variables’, which are analyzed within a machine learning framework rather than from a qualitative perspective. All lithological and pedological features that are relevant and available in an appropriate format (e.g. raster, shapefile) have already been incorporated into the model, independently from the classifications found in soil maps. Thus, we believe that integrating geological or soil maps would not contribute substantially to the analysis, since these variables are already considered in the model and are presented as part of the results, particularly in Figure 7, which already shows the spatial distribution of lithological, pedological, and land use data.

However, we agree that adding some further contextual information in the Study Area chapter could help clarify key aspects of the area's geomorphology. Therefore, we have revised the part referring to landslides as follows: “Landslides are the dominant land degradation process, exhibiting considerable variability in magnitude and frequency. In the study area, landslide activity is largely dominated by periodic reactivation of pre-existing large-scale landslide bodies, primarily triggered by intense or prolonged rainfall events. Fluvial undercutting at the toe of landslide deposits also plays a critical role in destabilizing valley slopes” (lines 153-157).

Regarding the suggestion to provide information on how the fields are ploughed, we unfortunately do not have detailed data on this. Most of the cultivated areas in the study area are classified as rainfed arable lands, with very few hectares of vineyards and orchards (they are not enough to justify a separate class as they are very small and sporadic polygons in the overall land use map). Please note that the variable ‘cultivated fields’ is among the feature variables integrated into the model, but the results did not reveal any significant relationship between this variable and the FTIR indices. Please also note that in the Study area chapter, we explicitly stated that: " Rainfed arable lands represent the predominant agricultural landscape in the watershed..." (line 168).

Reviewer_5: Table 3 is interesting, but the authors could include photographs of typical examples of rocks and a map of agricultural use of the research area.

CA: Thank you for your appreciation. In our opinion, adding photographs of representative rock types could be somewhat interesting, but it would not substantially add value to the research, as the characteristics of the lithological groups are already clearly described in Table 3. Moreover, this information is relevant to our study because it relates to the parent material of the soils. Additionally, the characteristics of sediments may be affected by the bedrock from which they come from, rather than to rock outcrops or lithological material themselves. Therefore, including pictures of rocks or outcrops may not be entirely relevant in this context. Regarding agricultural land use in the study area, a classification comprising 10 land use classes (including agricultural areas) has already been integrated into the model. A map showing the classified land use is also presented as part of the results in Figure 7.

Reviewer_6: After Fig. 5, the authors describe Figure 7 (line 439) in the text, so it should be moved so that the order of citation of graphics in the text is maintained. There is no description of Figure 6 in this paragraph. This needs to be supplemented. There is only one reference in line 473, but it is too little for the information contained there.

CA: Thank you for your comment. We understand the point you raised. However, Figure 6 was already cited at line 451 (previously line 436 in the former submitted version), before the citation of Figure 7 at line 454 (previously line 439). Therefore, the consistency in the figure numbering and citation order is maintained. Also, we would like to clarify that the textual information regarding basic statistics of the raster-based feature variables (Figures 6 and 7) is not reported in the manuscript text but only in the Supporting Information, as explicitly stated in the manuscript (lines 454-455). This was a specific request from the Editor during the initial revision round, before the manuscript was sent out for peer review. Indeed, the Editor specifically asked us to move all numerical and technical data (such as these basic statistics) to the Supporting Information, in order to avoid overloading the main text with non-essential content and to help readers focus on the most relevant information for interpretation and discussion. On the other hand, the subsequent information referring to Figure 7 (lines 456-467) mainly concerns the shapefile-based variables (i.e., lithology and land use), from which proportions were extracted. In this case, the text does not only present numerical data that could be moved to the Supporting Information, but also provides qualitative and descriptive insights into the distribution of these variables across the study area. These elements are indeed important to support the discussion (you will see that Figs. 7h and 7i have been extensively cited in the discussion).

Reviewer_7: Line 571, requires consideration, the hydrophobicity of matter alone does not have to stop erosion, which can be caused by, for example, soil instability and mechanical washing out. In addition, chemical compounds can appear in polluted waters that cause emulsification of such compounds.

CA: Thank you for your comment. However, in the discussion, we avoid any reference to a "protective" or "preventive" role of hydrophobicity against soil erosion. On the contrary, we stated the pedological significance of hydrophobicity on soil erosion in the Introduction, before outlining the objectives of the present work. We are fully aware that erosion can occur in areas affected by soil instability, which is the reason why we decided to focus on three different hydrological/geomorphological settings that take these aspects into account. All of this is addressed in the Discussion (see paragraphs below).

--- --- --- --- --- --- --- --- --- --- --- --- --- --- --- --- --- --- --- --- --- --- --- --- --- --- --- --- --- ---

Editor’s additional comments:

Editor_1: In addition to the reviewers' comments, I would like to draw your attention to several further issues that require careful consideration. While your integration of FTIR-derived indices with Random Forest modelling is methodologically sound, the interpretation of these proxies as direct indicators of erosion susceptibility appears overstated. The manuscript lacks empirical validation linking the FTIR indices to actual erosion measurements, sediment yields, or field-observed processes. This disconnect should be addressed with a more cautious interpretation of your findings, clearly acknowledging the proxy-based and indirect nature of your assessments.

CA: Thank you so much for the comment. We agree with your point of view. We have added a short paragraph in the section ‘5.3 Limitations of the methodological approach and improvements’: “In general, the significance of these findings relies on the assumption that the FTIR indices used in this study can serve as proxies for soil hydrophobicity, CEC, and organic matter–cation interactions (Table 1) within this environmental context, as supported by evidence drawn from the literature [39,50-55,84]. The obtained results represent an initial preliminary attempt towards a better understanding of such relationships. However, these results should be interpreted with caution, e.g., as we discussed for their relevance in terms of erosion processes. That is, an empirical evaluation linking FTIR indices to actual erosion measurements—necessary to establish these indices as direct indicators—is currently lacking. Nonetheless, additional studies are needed to explore more detailed relationships between these potential FTIR-based indicators and the chemical and pedological characteristics of sediment source areas” (Lines 737-745). Moreover, we would like to stress the fact that in the manuscript we are frequently referring to these indices as “potential indicators”, thereby implicitly reflecting their nature in relation to the methodology and the approach employed in the study. In any case, in the manuscript we never refer to definitive evidence or unquestionable results, but we consistently use modal verbs (e.g., could, might, may, etc…). In this regard, we have slightly refined some sentences (please check the Tracked Changes Copy).

Editor_2: Moreover, your conclusions frequently extrapolate results from a relatively small catchment in the Northern Apennines to “Mediterranean watersheds” in general. Such generalisations are not sufficiently substantiated and should be carefully delimited to the specific physiographic and climatic context of your study area. A more balanced formulation that frames your findings as case-study insights rather than general regional conclusions is recommended.

CA: Thank you for the valuable comment. We agree that, in the Conclusion, we should refer to these extrapolations more cautiously. Therefore, we have made some corrections:

- Lines 785-786: we have clearly specified that we selected a “temperate agricultural-forested area of the Northern Apennines chosen as representative of Mediterranean environments”;

- Lines 802-803: “This underscores the complexity of SOM dynamics in environments with similar characteristics and supports…”;

- Lines 810-812: “These findings carry significant implications for Mediterranean watersheds having similar vegetation, as well as for regions with seasonally contrasting climates and lithologies dominated by weak, stratified, clay-rich rocks”;

- Lines 816-818: “These insights suggest that such landscape-based modelling could be a valuable approach for predicting organic matter cycling and carbon sequestration in complex, dynamic Mediterranean watersheds with similar landscape configuration”.

Editor_3: Another aspect requiring attention concerns the sampling strategy. Given that samples were collected following a significant rainfall event, it would be pertinent to discuss how this may have influenced sediment composition and whether such cond

---

## [Editor Report · Decision Letter 1]

18 Jun 2025

PONE-D-25-04924R1Integration of FTIR-based soil degradation indices with stochastic modelling to assess spatial patterns of organic matter-sediment dynamics in a Mediterranean watershed – A case study from the Northern ApenninesPLOS ONE

Dear Dr. La Licata,

Thank you for submitting your manuscript to PLOS ONE. After careful consideration, we feel that it has merit but does not fully meet PLOS ONE’s publication criteria as it currently stands. Therefore, we invite you to submit a revised version of the manuscript that addresses the points raised during the review process.

The manuscript now fully meets PLOS ONE’s methodological and data requirements and addresses all major reviewer and editor concerns. A minor revision is requested solely to streamline the title and refine the keywords to avoid unnecessary repetition and ensure better indexing. No further substantive changes are required, and I recommend acceptance once these final editorial points have been implemented.

We look forward to receiving your revised manuscript.

Kind regards,

Przemysław Mroczek, Dr. hab.

Academic Editor

PLOS ONE

Journal Requirements:

Additional Editor Comments :

Dear Dr La Licata and Co-authors,

Thank you very much for submitting the revised version of your manuscript entitled “Integration of FTIR-based soil degradation indices with stochastic modelling to assess spatial patterns of organic matter–sediment dynamics in a Mediterranean watershed – A case study from the Northern Apennines” to PLOS ONE. I sincerely appreciate the significant effort you have devoted to addressing the reviewers’ and editorial comments. The manuscript has improved considerably in terms of methodological transparency, a more balanced interpretation of proxy-based findings, and an overall clearer presentation of the research context.

Before I can proceed with a final decision, I would kindly ask you to address two small but important editorial points to ensure that your manuscript meets the journal’s best practice for clarity and discoverability. Firstly, while your current title correctly reflects the study’s scope and regional focus, it is rather long and includes redundant wording that makes it unnecessarily complex. PLOS ONE encourages concise titles that convey the essential elements without superfluous phrases. I therefore strongly recommend shortening it to the following form: “FTIR-derived soil degradation indices and stochastic modelling of organic matter–sediment dynamics in a Mediterranean catchment: a Northern Apennines case study.” This version preserves all key terms but removes the overly extended structure and the phrase “to assess spatial patterns”, which is implicit in your methods and discussion.

Secondly, I encourage you to refine your set of keywords to avoid repeating phrases already present in the title. For example, “Organic matter indices”, “Mediterranean watershed”, and “Geomorphological modelling” duplicate core title terms and thus reduce the added value of the keywords for search engines and indexing. A more effective set could include terms that highlight important aspects of your study not directly stated in the title. I suggest the following as an example: Soil erosion risk, FTIR spectroscopy, Soil organic matter composition, Random Forest modelling, Watershed connectivity, and Sediment transport processes. This selection emphasises the main processes, analytical approach, and spatial system interactions central to your manuscript.

Once you have adjusted the title and keywords accordingly, I will be pleased to proceed with the final acceptance. No further substantive revisions are needed. Thank you again for your thoughtful and thorough response to the peer review process and for choosing PLOS ONE for the publication of your work. I look forward to receiving your final version soon.

With best regards,

Przemysław Mroczek, Dr hab.

---

## [Author Response · Author response to Decision Letter 2]

18 Jul 2025

Dear Editor,

We would like to thank you very much for all the work you have done with us in these months, during several revision processes, which has led to a result that we are very satisfied. Please find below our responses.

Editor’ comments:

Editor_1: Firstly, while your current title correctly reflects the study’s scope and regional focus, it is rather long and includes redundant wording that makes it unnecessarily complex. PLOS ONE encourages concise titles that convey the essential elements without superfluous phrases. I therefore strongly recommend shortening it to the following form: “FTIR-derived soil degradation indices and stochastic modelling of organic matter–sediment dynamics in a Mediterranean catchment: a Northern Apennines case study.” This version preserves all key terms but removes the overly extended structure and the phrase “to assess spatial patterns”, which is implicit in your methods and discussion.

Corresponding author (CA): Thank you very much for the comment and the suggestion. We have decided to adopt the title you proposed, with just one word changed. We chose to keep the term “watershed” instead of “catchment” for consistency with the terminology used in the text. The title now reads as follows: "FTIR-derived soil degradation indices and stochastic modelling of organic matter–sediment dynamics in a Mediterranean watershed: a Northern Apennines case study".

Editor_2: Secondly, I encourage you to refine your set of keywords to avoid repeating phrases already present in the title. For example, “Organic matter indices”, “Mediterranean watershed”, and “Geomorphological modelling” duplicate core title terms and thus reduce the added value of the keywords for search engines and indexing. A more effective set could include terms that highlight important aspects of your study not directly stated in the title. I suggest the following as an example: Soil erosion risk, FTIR spectroscopy, Soil organic matter composition, Random Forest modelling, Watershed connectivity, and Sediment transport processes. This selection emphasises the main processes, analytical approach, and spatial system interactions central to your manuscript.

CA: Thank you very much for your comment and suggestion. Also in this case, we decided to use essentially the keywords you proposed, with only minor adjustments. Now the keywords are: “Soil erosion susceptibility, FTIR spectroscopy, Soil organic matter composition, Random Forest modelling, Landscape connectivity, Sediment transport processes”. We chose to avoid the term “risk” because in applied geomorphology this term has a specific meaning that does not apply to the scope of our work. Therefore, we would prefer to avoid potential misunderstandings among specialized readers. Moreover, compared to your proposal, we decided to keep “landscape connectivity” instead of “watershed connectivity,” as this term (and its related definition) is more relevant and consistent with the literature cited in the manuscript.

---

## [Editor Report · Decision Letter 2]

30 Jul 2025

FTIR-derived soil degradation indices and stochastic modelling of organic matter–sediment dynamics in a Mediterranean watershed: a Northern Apennines case study

PONE-D-25-04924R2

Dear Dr. La Licata,

We’re pleased to inform you that your manuscript has been judged scientifically suitable for publication and will be formally accepted for publication once it meets all outstanding technical requirements.

Kind regards,

Przemysław Mroczek, Dr. hab.

Academic Editor

PLOS ONE

Additional Editor Comments (optional):

Thank you for your thorough revision and for addressing the editorial suggestions with care and clarity. The changes made to the title, keyword selection, and data availability statement are appropriate and fully in line with the previous guidance. I appreciate the effort to enhance the clarity and coherence of the manuscript.

However, I note that the revised version labelled as "tracked changes" does not actually contain any visible change markings. While the clean version appears to incorporate the necessary revisions, the absence of a tracked version prevents a straightforward verification of specific edits. I recommend ensuring that such a file is included in future submissions where applicable.

I would like to thank you and your co-authors for your constructive engagement throughout the editorial process. I believe the manuscript is now in a form suitable for publication.
---

## [Editor Report · Acceptance letter]

PONE-D-25-04924R2

PLOS ONE

Dear Dr. La Licata,

I'm pleased to inform you that your manuscript has been deemed suitable for publication in PLOS ONE. Congratulations! Your manuscript is now being handed over to our production team.

Kind regards,

on behalf of

Dr. hab. Przemysław Mroczek

Academic Editor

PLOS ONE